# *Presto!* Distilling Steps and Layers for Accelerating Music Generation

**Zachary Novack**[*]
UC – San Diego

**Ge Zhu & Jonah Casebeer**
Adobe Research

**Julian McAuley & Taylor Berg-Kirkpatrick**
UC – San Diego

**Nicholas J. Bryan**
Adobe Research

## Abstract

Despite advances in diffusion-based text-to-music (TTM) methods, efficient, high-quality generation remains a challenge. We introduce ***Presto!***, an approach to inference acceleration for score-based diffusion transformers via reducing both sampling steps and cost per step. To reduce steps, we develop a new score-based distribution matching distillation (DMD) method for the EDM-family of diffusion models, the first GAN-based distillation method for TTM. To reduce the cost per step, we develop a simple, but powerful improvement to a recent layer distillation method that improves learning via better preserving hidden state variance. Finally, we combine our step and layer distillation methods together for a dual-faceted approach. We evaluate our step and layer distillation methods independently and show each yield best-in-class performance. Our combined distillation method can generate high-quality outputs with improved diversity, accelerating our base model by 10-18x (230/435ms latency for 32 second mono/stereo 44.1kHz, 15x faster than the comparable SOTA model) — the fastest TTM to our knowledge.

## 1 Introduction

We have seen a renaissance of audio-domain generative media (Chen et al., 2024; Agostinelli et al., 2023; Liu et al., 2023; Copet et al., 2023), with increasing capabilities for both Text-to-Audio (TTA) and Text-to-Music (TTM) generation. This work has been driven in-part by audio-domain *diffusion models* (Song et al., 2020; Ho et al., 2020; Song et al., 2021), enabling considerably better audio modeling than generative adversarial network (GAN) or variational autoencoder (VAE) methods (Dhariwal & Nichol, 2021). Diffusion models, however, suffer from long inference times due to their iterative denoising process, requiring a substantial number of function evaluations (NFE) during inference (i.e. sampling) and resulting in ≈5-20 seconds at best for non-batched ≈32s outputs.

Accelerating diffusion inference typically focuses on *step distillation*, i.e. the process of reducing the *number* of sampling steps by distilling the diffusion model into a few-step generator. Methods include consistency-based (Salimans & Ho, 2022; Song et al., 2023; Kim et al., 2023) and adversarial (Sauer et al., 2023; Yin et al., 2023; 2024; Kang et al., 2024) approaches. Others have also investigated *layer-distillation* (Ma et al., 2024; Wimbauer et al., 2024; Moon et al., 2024), which draws from transformer early exiting (Hou et al., 2020; Schuster et al., 2021) by dropping interior layers to reduce the *cost* per sampling step for image generation. For TTA/TTM models, however, distillation techniques have only been applied to shorter or lower-quality audio (Bai et al., 2024; Novack et al., 2024a), necessitate ≈10 steps (vs. 1-4 step image methods) to match base quality (Saito et al., 2024), and have not successfully used layer or GAN-based distillation methods.

We present **Presto**[1], a dual-faceted distillation approach to inference acceleration for score-based diffusion transformers via reducing the number of sampling steps and the cost per step. **Presto** includes three distillation methods: (1) **Presto-S**, a new distribution matching distillation algorithm for *score-based*, EDM-style diffusion models (see Fig. 1) leveraging GAN-based step distillation

---

[*]Work done while an intern at Adobe.

[1]*Presto* is the common musical term denoting fast music from 168-200 beats per minute.

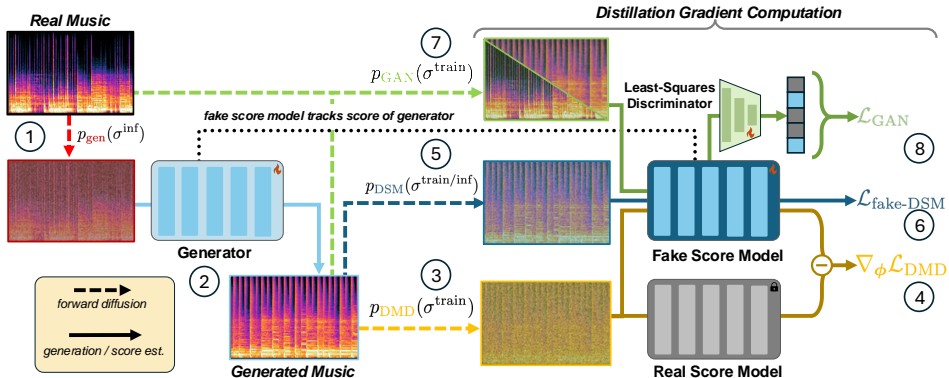

Figure 1: **Presto-S**. Our goal is to distill the initial "real" score model (grey) $\mu_{\boldsymbol{\theta}}$ into a few-step generator (light blue) $G_{\boldsymbol{\phi}}$ to minimize the KL divergence between the distribution of $G_{\boldsymbol{\phi}}$'s outputs and the real distribution. This requires that we train an auxiliary "fake" score model $\mu_{\boldsymbol{\psi}}$ (dark blue) that estimates the score of the *generator's* distribution at each gradient step. Formally: (1) real audio is corrupted with Gaussian noise sampled from the generator noise distribution $p_{\text{gen}}(\sigma^{\text{inf}})$ which is then (2) passed into the generator to get its output. Noise is then added to this generation according to three *different* noise distributions: (3) $p_{\text{DMD}}(\sigma^{\text{train}})$, which is (4) passed into both the real and fake score models to calculate the distribution matching gradient $\nabla_{\phi}\mathcal{L}_{\text{DMD}}$; (5) $p_{\text{DSM}}(\sigma^{\text{train/inf}})$, which is used to (6) train the fake score model on the *generator's* distribution with $\mathcal{L}_{\text{fake-DSM}}$; and (7) an adversarial distribution $p_{\text{GAN}}(\sigma^{\text{train}})$, which along with the real audio is (8) passed into a least-squares discriminator built on the fake score model's intermediate activations to calculate $\mathcal{L}_{\text{GAN}}$.

with the flexibility of *continuous-time* models, (2) **Presto-L**, a conditional layer distillation method designed to better preserve hidden state variance during distillation, and (3) **Presto-LS**, a combined layer-step distillation method that critically uses layer distillation and *then* step distillation while disentangling layer distillation from real and fake score-based gradient estimation.

To evaluate our approach, we ablate the design space for both distillation processes. First, we show our step distillation method achieves best-in-class acceleration and quality via careful choice of loss noise distributions, GAN design, and continuous-valued inputs, the first such method to match base TTM diffusion sampling quality with 4-step inference. Second, we show our layer distillation method offers a consistent improvement in both speed *and* performance over SOTA layer dropping methods and base diffusion sampling. Finally, we show that layer-step distillation accelerates our base model by 10-18x (230/435ms latency for 32 second mono/stereo 44.1kHz, 15x faster than the comparable SOTA model) while notably improving diversity over step-only distillation.

Overall, our core contributions include the development of a holistic approach to accelerating score-based diffusion transformers including : **(1)** The development of distribution matching distillation for continuous-time score-based diffusion (i.e. EDM), the first GAN-based distillation method for TTM. **(2)** The development of an improved layer distillation method that consistently improves upon both past layer distillation method and our base diffusion model. **(3)** The development of the first combined layer and step distillation method. **(4)** Evaluation showing our step, layer, and layer-step distillation methods are all best-in-class and, when combined, can accelerate our base model by 10-18x (230/435ms latency for 32 second mono/stereo 44.1kHz, 15x faster than Stable Audio Open (Evans et al., 2024c)), the fastest TTM model to our knowledge. For sound examples (anonymous link), see `https://presto-music.github.io/web/`.

## 2 BACKGROUND & RELATED WORK

### 2.1 MUSIC GENERATION

Audio-domain music generation methods commonly use autoregressive (AR) techniques (Zeghidour et al., 2021; Agostinelli et al., 2023; Copet et al., 2023) or diffusion (Forsgren & Martiros, 2022; Liu et al., 2023; 2024b; Schneider et al., 2023). Diffusion-based TTA/TTM (Forsgren & Martiros,

2022; Liu et al., 2023; 2024b; Schneider et al., 2023; Evans et al., 2024a) has shown the promise of full-text control (Huang et al., 2023), precise musical attribute control (Novack et al., 2024b;a; Tal et al., 2024), structured long-form generation (Evans et al., 2024b), and higher overall quality over AR methods (Evans et al., 2024a;b; Novack et al., 2024b; Evans et al., 2024c). The main downside of diffusion, however, is that it is slow and thus not amenable to interactive-rate control.

## 2.2 SCORE-BASED DIFFUSION MODELS

Continuous-time diffusion models have shown great promise over discrete-time models both for their improved performance on images (Balaji et al., 2022; Karras et al., 2023; Liu et al., 2024a) *and* audio (Nistal et al., 2024; Zhu et al., 2023; Saito et al., 2024), as well as their relationship to the general class of flow-based models (Sauer et al., 2024; Tal et al., 2024). Such models involve a forward noising process that gradually adds Gaussian noise to real audio signals $\boldsymbol{x}_{\text{real}}$ and a reverse process that transforms pure Gaussian noise back into data (Song et al., 2021; Sohl-Dickstein et al., 2015). The reverse process is defined by a stochastic differential equation (SDE) with an equivalent ordinary differential equation (ODE) form called the *probability flow* (PF) ODE (Song et al., 2021):

$$\mathrm{d}\boldsymbol{x} = -\sigma \nabla_{\boldsymbol{x}} \log p(\boldsymbol{x} \mid \sigma) \mathrm{d}\sigma, \qquad (1)$$

where $\nabla_{\boldsymbol{x}} \log p(\boldsymbol{x} \mid \sigma)$ is the score function of the marginal density of $\boldsymbol{x}$ (i.e. the noisy data) at noise level $\sigma$ according to the forward diffusion process. Thus, the goal of score-based diffusion models is to learn a *denoiser* network $\mu_{\boldsymbol{\theta}}$ such that $\mu_{\boldsymbol{\theta}}(\boldsymbol{x}, \sigma) = \mathbb{E}[\boldsymbol{x}_{\text{real}} \mid \boldsymbol{x}, \sigma]$. The score function is:

$$\nabla_{\boldsymbol{x}} \log p(\boldsymbol{x} \mid \sigma) \approx \frac{\boldsymbol{x} - \mu_{\boldsymbol{\theta}}(\boldsymbol{x}, \sigma)}{\sigma}. \qquad (2)$$

Given a trained score model, we can generate samples at inference time by setting a decreasing *noise schedule* of $N$ levels $\sigma_{\max} = \sigma_N > \sigma_{N-1} > \cdots > \sigma_0 = \sigma_{\min}$ and iteratively solving the ODE at these levels using our model and any off-the-shelf ODE solver (e.g. Euler, Heun).

The EDM-family (Karras et al., 2022; 2023) of score-based diffusion models is of particular interest and unifies several continuous-time model variants within a common framework and improves model parameterization and training process. The EDM score model is trained by minimizing a reweighted denoising score matching (DSM) loss (Song et al., 2021):

$$\mathcal{L}_{\text{DSM}} = \mathbb{E}_{\boldsymbol{x}_{\text{real}} \sim \mathcal{D}, \sigma \sim p(\sigma^{\text{train}}), \epsilon \sim \mathcal{N}(0, \boldsymbol{I})} \left[ \lambda(\sigma) \| \boldsymbol{x}_{\text{real}} - \mu_{\boldsymbol{\theta}}(\boldsymbol{x}_{\text{real}} + \epsilon\sigma, \sigma) \|_2^2 \right], \qquad (3)$$

where $p(\sigma^{\text{train}})$ denotes the *noise distribution* during training, and $\lambda(\sigma)$ is a noise-level weighting function. Notably, EDM defines a *different* noise distribution to discretize for inference $p(\sigma^{\text{inf}})$ that is distinct from $p(\sigma^{\text{train}})$ (see Fig. 2), as opposed to a noise schedule shared between training and inference. Additionally, EDMs represent the denoising network using extra noise-dependent preconditioning parameters, training a network $f_{\boldsymbol{\theta}}$ with the parameterization:

$$\mu_{\boldsymbol{\theta}}(\boldsymbol{x}, \sigma) = c_{\text{skip}}(\sigma)\boldsymbol{x} + c_{\text{out}}(\sigma) f_{\boldsymbol{\theta}}(c_{\text{in}}(\sigma)\boldsymbol{x}, c_{\text{noise}}(\sigma)). \qquad (4)$$

For TTM models, $\mu_{\boldsymbol{\theta}}$ is equipped with various condition embeddings (e.g. text) $\mu_{\boldsymbol{\theta}}(\boldsymbol{x}, \sigma, \boldsymbol{e})$. To increase text relevance and quality at the cost of diversity, we employ *classifier free guidance* (CFG) (Ho & Salimans, 2021), converting the denoised output to: $\tilde{\mu}_{\boldsymbol{\theta}}^w(\boldsymbol{x}, \sigma, \boldsymbol{e}) = \mu_{\boldsymbol{\theta}}(\boldsymbol{x}, \sigma, \emptyset) + w(\mu_{\boldsymbol{\theta}}(\boldsymbol{x}, \sigma, \boldsymbol{e}) - \mu_{\boldsymbol{\theta}}(\boldsymbol{x}, \sigma, \emptyset))$, where $w$ is the guidance weight and $\emptyset$ is a "null" conditioning.

## 2.3 DIFFUSION DISTILLATION

Step distillation is the process of reducing diffusion sampling steps by distilling a base model into a few-step generator. Such methods can be organized into two broad categories. Online consistency approaches such as consistency models (Song et al., 2023), consistency trajectory models (Kim et al., 2023), and variants (Ren et al., 2024; Wang et al., 2024a) distill directly by enforcing consistency across the diffusion trajectory and optionally include an adversarial loss (Kim et al., 2023). While such approaches have strong 1-step generation for images, attempts for audio have been less successful and only capable of generating short segment (i.e. $< 10$ seconds), applied to lower-quality base models limiting upper-bound performance, needing up to 16 sampling steps to match baseline quality (still slow), and/or did not successfully leverage adversarial losses which have been found to increase realism for other domains (Bai et al., 2024; Saito et al., 2024; Novack et al., 2024a).

In contrast, offline adversarial distillation methods include Diffusion2GAN (Kang et al., 2024), LADD (Sauer et al., 2024), and DMD (Yin et al., 2023). Such methods work by generating large amounts of offline noise–sample pairs from the base model, and finetuning the model into a conditional GAN for few-step synthesis. These methods can surpass their adversarial-free counterparts, yet require expensive offline data generation and massive compute infrastructure to be efficient.

Alternatively, improved DMD (DMD2) (Yin et al., 2024) introduces an online adversarial diffusion distillation method for images. DMD2 (1) removes the need for expensive offline data generation (2) adds a GAN loss and (3) outperforms consistency-based methods and improves overall quality. DMD2 primarily works by distilling a one- or few-step generator $G_\phi$ from a base diffusion model $\mu_{\text{real}}$, while simultaneously learning a score model of the generator's distribution online $\mu_{\text{fake}}$ in order to approximate a target KL objective (with $\mu_{\text{real}}$) used to train the generator. To our knowledge, there are no adversarial diffusion distillation methods for TTM or TTA.

Beyond step distillation, layer distillation, or the process of dropping interior layers to reduce the cost per sampling step, has been recently studied (Moon et al., 2024; Wimbauer et al., 2024). Layer distillation draws inspiration from transformer early exiting and layer caching (Hou et al., 2020; Schuster et al., 2021) and has found success for image diffusion, but has not been compared or combined with step distillation methods and has not been developed for TTA/TTM. In our work, we seek to understand how step and layer distillation interact for accelerating music generation.

## 3 *Presto!*

We propose a dual-faceted distillation approach for inference acceleration of continuous-time diffusion models. Continuous-time models have been shown to outperform discrete-time DDPM models (Song et al., 2020; Karras et al., 2022; 2024), but past DMD/DMD2 work focuses on the latter. Thus, we redefine DMD2 (a step distillation method) in Section 3.1 for continuous-time score models, then present an improved formulation and study its design space in Section 3.2. Second, we design a simple, but powerful improvement to the SOTA layer distillation method to understand the impact of reducing inference cost per step in Section 3.3. Finally, we investigate how to combine step and layer distillation methods together in Section 3.4.

### 3.1 EDM-Style Distribution Matching Distillation

We first redefine DMD2 in the language of continuous-time, score-based diffusion models (i.e. EDM-style). Our goal is to distill our score model $\mu_\theta$ (which we equivalently denote as $\mu_{\text{real}}$, as it is trained to model the score of real data) into an accelerated generator $G_\phi$ that can sample in 1-4 steps. Formally, we wish to minimize the reverse KL Divergence between the real distribution $p_{\text{real}}$ and the generator $G_\phi$'s distribution $p_{\text{fake}}$: $\mathcal{L}_{\text{DMD}} = D(p_{\text{fake}} \| p_{\text{real}})$. The KL term cannot be calculated explicitly, but we can calculate its *gradient* with respect to the generator if we can access the score of the generator's distribution. Thus, we also train a "fake" score model $\mu_\psi$ (or equivalently, $\mu_{\text{fake}}$) to approximate the generator distribution's *score function* at each gradient step during training.

First, given some real data $x_{\text{real}}$, we sample a noise level from a set of predefined levels $\sigma \sim \{\sigma_i\}_{\text{gen}}$, and then pass the corrupted real data through the generator to get the generated output $\hat{x}_{\text{gen}} = G_\phi(x_{\text{real}} + \sigma\epsilon, \sigma)$, where $\epsilon \sim \mathcal{N}(0, I)$ (we omit the conditioning $e$ for brevity). The gradient of the KL divergence between the real and the generator's distribution can then be calculated as:

$$\nabla_\phi \mathcal{L}_{\text{DMD}} = \mathbb{E}_{\sigma \sim \{\sigma_i\}, \epsilon \sim \mathcal{N}(0,I)} \left[ \left( \left( \mu_{\text{fake}}(\hat{x}_{\text{gen}} + \sigma\epsilon, \sigma) - \tilde{\mu}_{\text{real}}^w(\hat{x}_{\text{gen}} + \sigma\epsilon, \sigma) \right) \nabla_\phi \hat{x}_{\text{gen}} \right], \quad (5)$$

where $\{\sigma_i\}$ are the predefined noise levels for all loss calculations, and $\tilde{\mu}_{\text{real}}^w$ is the *CFG-augmented* real score model. To ensure that $\mu_{\text{fake}}$ accurately models the score of the generator's distribution at each gradient update, we train the fake score model with the weighted-DSM loss (i.e. standard diffusion training), but on *the generator outputs*:

$$\arg \min_\psi \mathcal{L}_{\text{fake-DSM}} = \mathbb{E}_{\sigma \sim \{\sigma_i\}, \epsilon \sim \mathcal{N}(0,I)} \left[ \lambda(\sigma) \| \hat{x}_{\text{gen}} - \mu_{\text{fake}}(\hat{x}_{\text{gen}} + \sigma\epsilon, \sigma) \|_2^2 \right] \quad (6)$$

To avoid using offline data (Yin et al., 2023), the fake score model is updated *5 times as often* as the generator to stabilize the estimation of the generator's distribution. DMD2 additionally includes an explicit adversarial loss in order to improve quality. Specifically, a discriminator head $D_\psi$ is

attached to the intermediate feature activations of the fake score network $\mu_{\text{fake}}$, and thus is trained with the nonsaturating GAN loss:

$$\arg\min_{\phi}\max_{\psi}\mathbb{E}_{\substack{\sigma\sim\{\sigma_i\},\\\epsilon\sim\mathcal{N}(0,\boldsymbol{I})}}\left[\log D_{\boldsymbol{\psi}}(\boldsymbol{x}_{\text{real}}+\sigma\epsilon,\sigma)\right]+\mathbb{E}_{\substack{\sigma\sim\{\sigma_i\},\\\epsilon\sim\mathcal{N}(0,\boldsymbol{I})}}\left[-\log D_{\boldsymbol{\psi}}(\hat{\boldsymbol{x}}_{\text{gen}}+\sigma\epsilon,\sigma)\right], \quad (7)$$

which follows past work on using diffusion model backbones as discriminators (Sauer et al., 2024). In all, the generator $G_{\phi}$ is thus trained with a combination of the distribution matching loss $\mathcal{L}_{\text{DMD}}$ and the adversarial loss $\mathcal{L}_{\text{GAN}}$, while the fake score model (and its discriminator head) is trained with the fake DSM loss $\mathcal{L}_{\text{fake-DSM}}$ and the adversarial loss $\mathcal{L}_{\text{GAN}}$. To sample from the distilled generator, DMD2 uses consistency model-style "ping-pong sampling" (Song et al., 2023), where the model iteratively denoises (starting at pure noise $\sigma_{\text{max}}$) and *renoises* to progressively smaller noise levels.

Regarding past work, we note Yin et al. (2024) *did* present a small-scale EDM-style experiment, but treated EDM models as if they were functions of discrete noise timesteps. This re-discretization runs counterintuitive to using score-based models for distribution matching, since the fake and real score models are meant to be run and trained in continuous-time and can adapt to variable points along the noise process. Furthermore, this disregards the ability of continuous-time models to capture the *entire* noise process from noise to data and enable *exact* likelihoods rather than ELBOs (Song et al., 2021). Additionally, since DMD2 implicitly aims to learn an integrator of the PF ODE $G_{\phi}(\boldsymbol{x},\sigma) \approx \boldsymbol{x} + \int_{\sigma}^{\sigma_{\text{min}}} -\delta\nabla\log p(\boldsymbol{x}\mid\delta)\mathrm{d}\delta$ (like other data-prediction distillation methods (Song et al., 2023)), learning this integral for any small set $\{\sigma_i\}$ restricts the generator's modeling capacity.

## 3.2 PRESTO-S: SCORE-BASED DISTRIBUTION MATCHING DISTILLATION

We develop our *score-based* distribution matching step distillation, **Presto-S** below and in Fig. 1 as well as the algorithm in Appendix A.3, a pseudo-code walkthrough in Appendix A.4, and expanded visualization in Appendix A.5.

### 3.2.1 CONTINUOUS-TIME GENERATOR INPUTS

In Section 3.1, the noise level and/or timestep is sampled from a *discrete*, hand-chosen set $\{\sigma_i\}_{\text{gen}}$. Discretizing inputs, however, forces the model to 1) be a function of a specific *number* of steps, requiring users to retrain separate models for each desired step budget (Yin et al., 2024; Kohler et al., 2024) and 2) be a function of *specific* noise levels, which may not be optimally aligned with where different structural, semantic, and perceptual features arise in the diffusion process (Si et al., 2024; Kynkäänniemi et al., 2024; Balaji et al., 2022; Sabour et al., 2024). When extending to continuous-time models, we train the distilled generator $G_{\phi}$ as a function of the continuous noise level sampled from the *distribution* $\sigma \sim p(\sigma)$. This allows our generator to both adapt better to variable budgets and to variable noise levels, as the generator can be trained with all noise levels sampled from $p(\sigma)$.

### 3.2.2 PERCEPTUAL LOSS WEIGHTING WITH VARIABLE NOISE DISTRIBUTIONS

A key difference between discrete-time and continuous-time diffusion models is the need for *discretization* of the noise process during inference. In discrete models, a single noise schedule defines a particular mapping between timestep $t$ and its noise level $\sigma$, and is fixed throughout training and inference. In continuous-time EDM models, however, we use a noise *distribution* $p(\sigma^{\text{train}})$ to sample during training, and a separate noise distribution for inference $p(\sigma^{\text{inf}})$ that is discretized to define the sampling schedule. In particular, when viewed in terms of the *signal-to-noise ratio* $1/\sigma^2$ or SNR as shown in Fig. 2, the *training* noise distribution puts

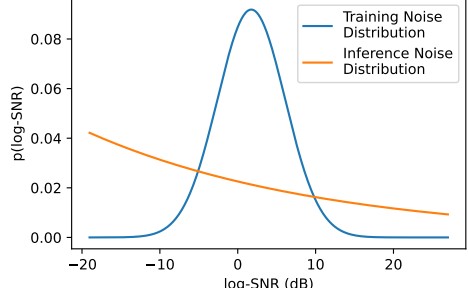

Figure 2: Training/Inference distributions for EDM models, in decibel SNR space.

the majority of its mass in the mid-to-high SNR range of the diffusion process. This design choice focuses on semantic and perceptual features, while the *inference* noise distribution is more evenly distributed but with a bias towards the low-SNR region, giving a bias to low-frequency features.

However, recall that *every* loss term including (5), (6), and (7) requires an additional re-corruption process that must follow a noise distribution, significantly expanding the design space for score-based models. Thus, we disentangle these forward diffusion processes and replace the shared discrete noise set with four *separate noise distributions* $p_{\text{gen}}$, $p_{\text{DMD}}$, $p_{\text{DSM}}$, and $p_{\text{GAN}}$, corresponding to the inputs to the generator and each loss term respectively, with no restrictions on how each weights each noise level (rather than forcing a particular noise weighting for all computation).

Then, if we apply the original DMD2 method naively to the EDM-style of score-models, we get $p_{\text{gen}}(\sigma^{\text{inf}}) = p_{\text{DMD}}(\sigma^{\text{inf}}) = p_{\text{DSM}}(\sigma^{\text{inf}}) = p_{\text{GAN}}(\sigma^{\text{inf}})$. This choice of $p_{\text{gen}}(\sigma^{\text{inf}})$ reasonably aligns the generator inputs during distillation to the inference process itself, but each loss noise distribution is somewhat misaligned from its role in the distillation process. In particular:

- $p_{\text{DMD}}$: The distribution matching gradient is the only point that the generator gets a signal from the *CFG-augmented* outputs of the teacher. CFG is critical for text following, but *primarily* within the mid-to-high SNR region of the noise (Kynkäänniemi et al., 2024).

- $p_{\text{GAN}}$: As in most adversarial distillation methods (Sauer et al., 2023; Yin et al., 2023), the adversarial loss's main strength is to increase the perceptual *realism/quality* of the outputs, which arise in the mid-to-high SNR regions, rather than structural elements.

- $p_{\text{DSM}}$: The score model training should in theory mimic standard diffusion training, and may benefit from the training distribution's provably faster convergence (Wang et al., 2024b) (as the fake score model is updated *online* to track the generator's distribution).

Thus, we shift all of the above terms to use the training distribution $p_{\text{DMD}}(\sigma^{\text{train}})$, $p_{\text{DSM}}(\sigma^{\text{train}})$ and $p_{\text{GAN}}(\sigma^{\text{train}})$ to force the distillation process to focus on perceptually relevant noise regions.

### 3.2.3 AUDIO-ALIGNED DISCRIMINATOR DESIGN

The original DMD2 uses a classic non-saturating GAN loss. The discriminator is a series of convolutional blocks downsampling the intermediate features into a *single* probability for real vs. fake. While this approach is standard in image-domain applications, many recent adversarial waveform synthesis works (Kumar et al., 2023; Zhu et al., 2024) use a *Least-Squares* GAN loss:

$$\arg \min_{\phi} \max_{\psi} \mathbb{E}_{\substack{\sigma \sim p_{\text{GAN}}(\sigma^{\text{train}}),\\ \epsilon \sim \mathcal{N}(0,\boldsymbol{I})}} [\|D_{\boldsymbol{\psi}}(\boldsymbol{x}_{\text{real}} + \sigma\epsilon, \sigma)\|_2^2] + \mathbb{E}_{\substack{\sigma \sim p_{\text{GAN}}(\sigma^{\text{train}}),\\ \epsilon \sim \mathcal{N}(0,\boldsymbol{I})}} [\|1 - D_{\boldsymbol{\psi}}(\hat{\boldsymbol{x}}_{\text{gen}} + \sigma\epsilon, \sigma)\|_2^2], \quad (8)$$

where the outputs of the discriminator $D_{\boldsymbol{\psi}}$ are only *partially* downsampled into a lower-resolution version of the input data (in this case, a latent 1-D tensor). This forces the discriminator to attend to more fine-grained, temporally-aligned features for determining realness, as the loss is averaged across the partially downsampled discriminator outputs. Hence, we use this style of discriminator for **Presto-S** to both improve and stabilize (Mao et al., 2017) the GAN gradient into our generator.

### 3.3 PRESTO-L: VARIANCE AND BUDGET-AWARE LAYER DROPPING

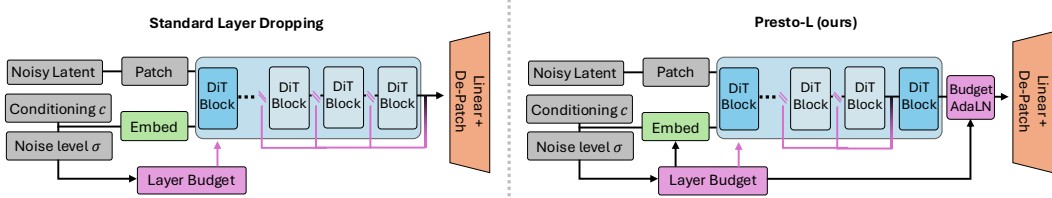

Figure 3: Baseline layer dropping (left) vs. **Presto-L** (right). Standard layer dropping uses the noise level $\sigma$ to set the budget of layers to drop, starting from the back of the DiT blocks. **Presto-L** shifts this dropping by one to the second-to-last block and adds explicit budget conditioning.

Given our step distillation approach above, we now seek to reduce the cost of individual *steps* themselves through layer distillation, and then combine both step and layer distillation in Section 3.4. We begin with the current SOTA method: ASE (Moon et al., 2024). ASE employs a fixed dropping schedule that monotonically maps noise levels to compute budgets, allocating more layers to lower noise levels. We enhance this method in three key ways: (1) ensuring consistent variance in layer distilled outputs, (2) implementing explicit budget conditioning, and (3) aligning layer-dropped outputs through direct distillation. See Appendix A.6 for more details.

**Variance Preservation**: First, we inspect the within-layer activation variance of our base model in Fig. 4. We find that while the variance predictably increases over depth, it notably spikes *on the last layer* up to an order of magnitude higher, indicating that the last layer behaves much differently as it is the direct input to the linear de-embedding layer. ASE, however, always drops layers starting from the *last* layer and working backwards, thus always removing this behavior. Hence, we remedy this fact and *shift* the layer dropping schedule by 1 to drop starting at the *second* to last layer, always rerouting back into the final layer to preserve the final layer's behavior.

**Budget Conditioning**: We include *explicit* budget conditioning into the model itself so that the model can directly adapt computation to the budget level. This conditioning comes in two places: (1) a global budget embedding added to the noise level embedding, thus contributing to the internal Adaptive Layer Norm (AdaLN) conditioning inside the DiT blocks, and (2) an additional AdaLN layer on the outset of the DiT blocks conditional only on the budget, in order to effectively rescale the outputs to account for the change in variance. Following (Peebles & Xie, 2023; Zhang et al., 2023), we zero-initialize both budget conditioning modules to improve finetuning stability.

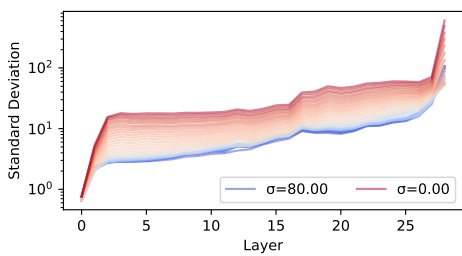

Figure 4: Hidden activation variance vs. layer depth. Each line is a unique noise level.

**Knowledge Distillation**: To encourage distillation without holding the base model in memory, we employ a *self-teacher* loss. Formally, if $h_L(\boldsymbol{x}, \boldsymbol{\theta})$ and $h_{\text{full}}(\boldsymbol{x}, \boldsymbol{\theta})$ are the normalized outputs of the final DiT layer with and without layer dropping respectively, the self-teacher loss is $\mathcal{L}_{\text{st}} = \|h_L(\boldsymbol{x}, \boldsymbol{\theta}) - \text{sg}(h_{\text{full}}(\boldsymbol{x}, \boldsymbol{\theta}))\|_2^2$, where $\text{sg}$ denotes a stop-gradient. This gives additional supervision during the early phases of finetuning so the layer-dropped outputs can match full model performance.

We show the differences between our **Presto-L** and the baseline approach in Fig. 3. By conditioning directly on the budget, and shifting the dropping schedule to account for the final DiT block behavior, we able to more adapt computation for reduced budgets while preserving performance.

## 3.4 Presto-LS: Layer-Step Distillation

As the act of layer distillation is, in principle, unrelated to the step distillation, there is no reason *a priori* that these methods could not work together. However, we found combining such methods to be surprisingly non-trivial. In particular, we empirically find that attempting both performing **Presto-L** finetuning and **Presto-S** at the same time OR performing **Presto-L** finetuning from an initial **Presto-S** checkpoint results in large instability and model degradation, as the discriminator dominates the optimization process and achieves near-perfect accuracy on real data.

We instead find three key factors in making combined step and layer distillation work: (1) *Layer-Step Distillation* – we first perform layer distillation then step distillation, which is more stable as the already-finetuned layer dropping prevents generator collapse; (2) *Full Capacity Score Estimation* – we keep the real and fake score models initialized from the *original* score model rather than the layer-distilled model, as this stabilizes the distribution matching gradient and provides regularization to the discriminator since the fake score model and the generator are initialized with different weights; and (3) *Reduced Dropping Budget* – we keep more layers during the layer distillation. We discuss more in Section 4.6 and how alternatives fail in Appendix A.7.

## 4 Experiments

We show the efficacy of **Presto** via a number of experiments. We first ablate the design choices afforded by **Presto-S**, and separately show how **Presto-L** flatly improves standard diffusion sampling. We then show how **Presto-L** and **Presto-S** stack up against SOTA baselines, and how we can combine such approaches for further acceleration, with both quantitative and subjective metrics. We finish by describing a number of extensions enabled by our accelerated, continuous-time framework.

## 4.1 SETUP

**Model:** We use latent diffusion (Rombach et al., 2022) with a fully convolutional VAE (Kumar et al., 2023) to generate mono 44.1kHz audio and convert to stereo using MusicHiFi (Zhu et al., 2024). Our latent diffusion model builds upon DiT-XL (Peebles & Xie, 2023) and takes in three conditioning signals: the noise level, text prompts, and beat per minute (BPM) for each song. We use FlashAttention-2 (Dao, 2023) for the DiT and `torch.compile` for the VAE decoder and MusicHiFi. For more details, see Appendix A.1.

**Data:** We use a 3.6K hour dataset of mono 44.1 kHz licensed instrumental music, augmented with pitch-shifting and time-stretching. Data includes musical meta-data and synthetic captions. For evaluation, we use Song Describer (no vocals) (Manco et al., 2023) split into 32 second chunks.

**Baselines:** We compare against a number of acceleration algorithms using our base model: Consistency Models (CM) (Song et al., 2023), SoundCTM (Saito et al., 2024), DITTO-CTM (Novack et al., 2024a), DMD-GAN (Yin et al., 2024), and ASE (Moon et al., 2024), as well as MusicGen (Copet et al., 2023) and Stable Audio Open (Evans et al., 2024c). See Appendix A.2 for more details.

**Metrics:** We use a number of common evaluation metrics for text-to-music generation, including distributional quality/diversity metrics (FAD/MMD/Density/Recall/Coverage), prompt adherence (CLAP Score), and latency (RTF). See Appendix A.2 for more details.

## 4.2 EXPLORING THE DESIGN SPACE OF **PRESTO-S**

**Loss Distribution Choice:** In Table 1 (Top), we show the FAD, MMD, and CLAP score for many **Presto-S** distilled models with different noise distribution choices. We find that the original DMD2 (Yin et al., 2024) setup (first row) underperforms compared to adapting the loss distributions. The largest change is in switching $p_{DMD}$ to the training distribution, which improves all metrics. This confirms our hypothesis that by focusing on the region most important for text guidance (Kynkäänniemi et al., 2024), we improve both audio quality and text adherence. Switching $p_{GAN}$ to the training

| $p_{gen}$ | $p_{DMD}$ | $p_{DSM}$ | $p_{GAN}$ | FAD | MMD | CLAP |
|---|---|---|---|---|---|---|
| **Least-Squares GAN** | | | | | | |
| Inf. | Inf. | Inf. | Inf. | 0.37 | 1.73 | 27.45 |
| Inf. | Inf. | Tr. | Inf. | 0.37 | 1.58 | 26.45 |
| Inf. | Inf. | Tr. | Tr. | 0.37 | 1.51 | 24.90 |
| Inf. | Tr. | Tr. | Inf. | 0.27 | 1.27 | 33.12 |
| Inf. | Tr. | Inf. | Tr. | 0.23 | 0.86 | **33.29** |
| Inf. | Tr. | Tr. | Tr. | **0.22** | **0.83** | 33.13 |
| Tr. | Tr. | Tr. | Tr. | 0.24 | 0.99 | 30.89 |
| **Non-Saturating GAN** | | | | | | |
| Inf. | Tr. | Inf. | Tr. | 0.24 | 0.89 | 31.48 |
| Inf. | Tr. | Tr. | Tr. | 0.25 | 0.96 | 31.78 |
| Tr. | Tr. | Tr. | Tr. | 0.26 | 1.04 | 29.46 |

Table 1: (Top) Comparing different choices of noise distribution for the **Presto-S** process. (Bottom) for best performing noise distributions, performance for standard GAN design vs. proposed least-squares GAN.

distribution also helps; in this case, the discriminator is made to focus on higher-frequency features (Si et al., 2024), benefiting quality. We also find only a small improvement when using the training distribution for $p_{DSM}$. This suggests that while the training distribution should lead to more stable learning of the online generator's score (Wang et al., 2024b), this may not be crucial. For all remaining experiments, we use $p_{DMD}(\sigma^{train}) = p_{GAN}(\sigma^{train}) = p_{DSM}(\sigma^{train})$ and $p_{gen}(\sigma^{inf})$.

**Discriminator Design:** We ablate the effect of switching from the chosen least-squares discriminator to the original softplus non-saturating discriminator, which notable treats the discriminator as a binary classifier and predicts the probability of real/generated. In Table 1 (Bottom), we find that using the least-squares discriminator leads to consistent improvements in audio quality (FAD/MMD) and in particular text relevance (CLAP), owing to the increased stability from the least-squares GAN.

**Continuous vs. Discrete Generator Inputs:** We test how *continuous-time* conditioning compares against a discrete and find the former is preferred as shown in Fig. 5. Continuous noise levels maintain a correlation where more steps improve quality, while discrete time models are more inconsistent. Additionally, the continuous-time conditioning performs best in text relevance. While the 1 and 2-step discrete models show slightly better FAD metrics than continuous on 1 and 2-step sampling, these models have a failure mode as shown in Fig. 13: 2-step discrete models drop high-frequency information and render transients (i.e. drum hits) poorly for genres like R&B or hip-hop.

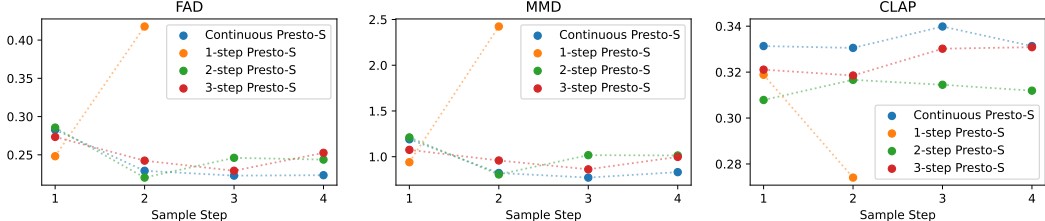

Figure 5: Continuous generator inputs vs. discrete inputs. Continuous inputs shows more consistent scaling with compute, while generally performing better in both quality and text relevance.

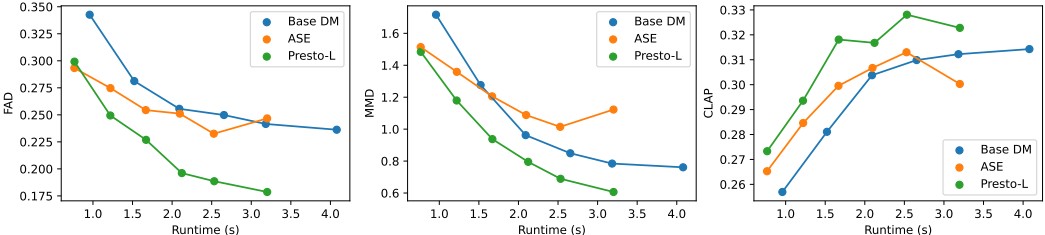

Figure 6: **Presto-L** results. **Presto-L** improves both the latency *and* the overall performance across all metrics, against both the leading layer dropping baseline and the base model.

### 4.3 PRESTO-L RESULTS

We compare **Presto-L** with both our baseline diffusion model and ASE (Moon et al., 2024) using the 2nd order DPM++ sampler (Lu et al., 2022) with CFG++ (Chung et al., 2024). For ASE and **Presto-L**, we use the optimal "D3" configuration from Moon et al. (2024), which corresponds to a dropping schedule, in terms of decreasing noise level (in quintiles), of $[14, 12, 8, 4, 0]$ (i.e. we drop 14 layers for noise levels in the top quintile, 12 for the next highest quintile, and so on). Layer distillation results at various sampling budgets are shown in Fig. 6. **Presto-L** yields an improvement over the base model on all metrics, speeding up by $\approx 27\%$ *and* improving quality and text relevance. ASE provides similar acceleration but degrades performance at high sampling steps and scales inconsistently. Dropping layers *improving* performance can be viewed via the lens of multi-task learning, where (1) denoising each noise level is a different task (2) later layers only activating for lower noise levels enables specialization for higher frequencies. See Appendix A.10 for further ablations.

### 4.4 FULL COMPARISON

In Table 2, we compare against multiple baselines and external models. For step distillation, **Presto-S** is best-in-class and the only distillation method to close to base model quality, while achieving an over 15x speedup in RTF from the base model. Additionally, **Presto-LS** improves performance for MMD, beating the base model with further speedups (230/435ms latency for 32 second mono/stereo 44.1kHz on an A100 40 GB). We also find **Presto-LS** improves *diversity* with higher recall. Overall, **Presto-LS** is 15x faster than SAO. We investigate latency more in Appendix A.9.

### 4.5 LISTENING TEST

We also conducted a subjective listening test to compare **Presto-LS** with our base model, the best non-adversarial distillation technique SoundCTM (Saito et al., 2024) distilled from our base model, and Stable Audio Open (Evans et al., 2024c). Users ($n = 16$) were given 20 sets of examples generated from each model (randomly cut to 10s for brevity) using random prompts from Song Describer and asked to rate the musical quality, taking into account both fidelity and semantic text match between 0-100. We run multiple paired t-tests with Bonferroni correction and find **Presto-LS** rates highest against all baselines ($p < 0.05$). We show additional plots in Fig. 14.

### 4.6 PRESTO-LS QUALITATIVE ANALYSIS

While **Presto-LS** improves speed and quality/diversity over step-only distillation, the increases are modest, as the dropping schedule for **Presto-L** was reduced ($[12, 8, 8, 0, 0]$) for step distillation

| Model | NFE | RTF-M/S (↑) | FAD (↓) | MMD (↓) | CLAP Score (↑) | Density (↑) | Recall (↑) | Coverage(↑) |
|---|---|---|---|---|---|---|---|---|
| **External Baselines*** | | | | | | | | |
| MusicGen-Small | 1.6K | 0.77 | 0.31 | 1.60 | 30.61 | 0.36 | 0.16 | 0.43 |
| MusicGen-Medium | 1.6K | 0.39 | 0.27 | 1.30 | 31.85 | 0.43 | 0.19 | 0.54 |
| MusicGen-Large | 1.6K | 0.37 | 0.25 | 1.21 | 32.83 | 0.44 | 0.15 | 0.54 |
| Stable Audio Open | 100 | 4.54 | 0.23 | 1.07 | 35.05 | 0.29 | 0.37 | 0.49 |
| **Base Model, Diffusion Sampling** | | | | | | | | |
| DPM-2S | 80 | 7.72 / 7.34 | 0.24 | 0.82 | 31.56 | 0.31 | 0.20 | 0.41 |
| DPM-2S+ASE | 80 | 9.80 / 9.22 | 0.25 | 1.12 | 30.03 | 0.27 | 0.16 | 0.41 |
| DPM-2S+**Presto-L** (ours) | 80 | 9.80 / 9.22 | 0.18 | 0.61 | 32.28 | 0.38 | 0.29 | 0.51 |
| **Consistency-Based Distillation** | | | | | | | | |
| CM | 4 | 118.77 / 67.41 | 0.47 | 2.50 | 26.33 | 0.17 | 0.01 | 0.16 |
| SoundCTM | 4 | 105.78 / 63.01 | 0.35 | 1.72 | 29.61 | 0.17 | 0.17 | 0.26 |
| DITTO-CTM | 4 | 118.77 / 67.41 | 0.36 | 1.62 | 28.31 | 0.22 | 0.04 | 0.32 |
| **Adversarial Distillation** | | | | | | | | |
| DMD-GAN | 4 | 118.77 / 67.41 | 0.29 | 1.16 | 27.56 | 0.57 | 0.07 | 0.41 |
| **Presto-S** (ours) | 4 | 118.77 / 67.41 | 0.22 | 0.83 | 33.13 | 0.60 | 0.10 | 0.50 |
| **Presto-LS** (ours) | 4 | 138.84 / 73.43 | 0.23 | 0.73 | 32.21 | 0.49 | 0.14 | 0.48 |

Table 2: Full Results on Song Describer (No vocals).*External baseline RTFs are all natively stereo.

stability. To investigate more, we analyze the hidden state activation variance of our step-distilled model in Fig. 7. The behavior is quite different than the base model, as the "spike" in the final layer is more amortized across the last 10 layers and never reaches the base model's magnitude. We hypothesize step-distilled models have more unique computation *throughout* each DiT block, making layer dropping difficult.

### 4.7 EXTENSIONS

**Adaptive Step Schedule:** A benefit of our continuous-time distillation is that besides setting how many steps (e.g., 1-4), we can set *where* those steps occur along the diffusion process by tuning the $\rho$ parameter in the EDM inference schedule, which is normally set to $\rho = 7$. In particular, decreasing $\rho$ (lower bounded by 1) puts more weight on low-SNR features and increasing $\rho$ on higher-SNR features (Karras et al., 2022). Qualitatively, we find that this process enables increased diversity of outputs, even from the same latent code (see Appendix A.8).

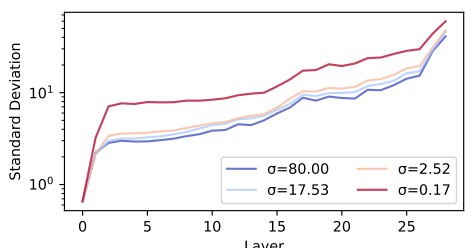

Figure 7: **Presto-S** hidden activation var.

**CPU Runtime:** We benchmark **Presto-LS**'s speed performance for CPU inference. On an Intel Xeon Platinum 8275CL CPU, we achieve a mono RTF of 0.74, generating 32 seconds of audio in 43.34 seconds. We hope to explore further CPU acceleration in future work.

**Fast Inference-Time Rejection Sampling:** Given **Presto-LS**'s speed, we investigated using *inference-time* compute to improve performance. Formally, we test the idea of *rejection sampling*, inspired by Kim et al. (2023), where we generate a batch of samples and reject $r$ fraction of them according to some ranking function. We use the CLAP score to discard samples that have poor text relevance. Over a number of rejection ratios (see Fig. 15), we find that CLAP rejection sampling strongly improves text relevance while maintaining or *improving* quality at the cost of diversity.

### 5 CONCLUSION

We proposed **Presto**, a dual-faceted approach to accelerating latent diffusion transformers by reducing sampling steps and cost per step via distillation. Our core contributions include the development of score-based distribution matching distillation (the first GAN-based distillation for TTM), a new layer distillation method, the first combined layer-step distillation, and evaluation showing each method are independently best-in-class and, when combined, can accelerate our base model by 10-18x (230/435ms latency for 32 second mono/stereo 44.1kHz, 15x faster than the comparable SOTA model), resulting in the fastest TTM model to our knowledge. We hope our work will motivate continued work on (1) fusing step and layer distillation and (2) new distillation of methods for continuous-time score models across media modalities such as image and video.

ACKNOWLEDGEMENTS

We would like to thank Juan-Pablo Caceres, Hanieh Deilamsalehy, and Chinmay Talegaonkar.

ETHICS STATEMENT AND REPRODUCIBILITY

As TTM systems become more powerful, there is both the opportunity to increase accessibility of musical expression, but also concern such systems may compete with creators. To reduce risk, we train our TTM work only on instrumental *licensed* music. Additionally, we hope that our focus on efficiency is useful to eventually make interactive-rate co-creation tools, allowing for greater flexibility and faster ideation. Following these concerns, we do not plan to release our model, but have done our best to compare against multiple open source baselines and/or re-train alternative methods for comparison and in-depth understanding of the reproducible insights of our work.

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

# A APPENDIX

## A.1 MODEL DESIGN DETAILS

As we perform latent diffusion, we first train a variational autoencoder. We build on the Improved RVQGAN (Kumar et al., 2023) architecture and training scheme by using a KL-bottleneck with a dimension of 32 and an effective hop of 960 samples, resulting in an approximately 45 Hz VAE. We train to convergence using the recommended mel-reconstruction loss and the least-squares GAN formulation with L1 feature matching on multi-period and multi-band discriminators.

Our proposed base score model backbone builds upon DiT-XL (Peebles & Xie, 2023), with modifications aimed at optimizing computational efficiency. Specifically, we use a streamlined transformer block design, consisting of a single attention layer followed by a single feed-forward layer, similar to Llama (Dubey et al., 2024). Our model utilizes three types of conditions including noise levels (timesteps) for score estimation, beat per minute (BPM) values of the song, and text descriptions. Following EDM, we apply a logarithmic transformation to the noise levels, followed by sinusoidal embeddings. Similarly, BPM values are input as scalars then go through sinusoidal embeddings to generate BPM embeddings. These processed noise-level embeddings and BPM embeddings are then combined and integrated into the DiT block through an adaptive layer normalization block. For text conditioning, we compute text embedding tokens with T5-based encoders and concatenate with audio tokens at each attention layer. As a result, the audio token query attends to a concatenated sequence of audio and text keys, enabling the model to jointly extract relevant information from both modalities. To provide baseline architectural speedups, we use FlashAttention-2 (Dao, 2023) for the DiT and Pytorch 2.0's built in graph compilation (Ansel et al., 2024) for the VAE decoder and MusicHifi mono-to-stereo.

Our discriminator design follows Yin et al. (2024) with a number of small modifications. $D_\psi$ consists of 4 blocks of 1D convolutions interleaved with GroupNorm and SiLU activations, and a final linear layer to collapse the channel dimension. The discriminator thus does not use any final linear layer to project to a single value, and instead its' output is *also* a 1D sequence but at even heavier downsampling than the input representation at $\approx$2.8 Hz. The discriminator receives its' input from the output of the 14th DiT Block (i.e. the halfway point through our 28 block DiT), as DiTs lack a clear "bottleneck" layer to place the discriminator like in UNets. We leave further investigation into discriminator design and placement inside the model for future work.

For the diffusion model hyparameter design, we follow Karras et al. (2024). Specifically, we set $\sigma_{\text{data}} = 0.5$, $P_{\text{mean}} = -0.4$, $P_{\text{std}} = 1.0$, $\sigma_{\text{max}} = 80$, $\sigma_{\text{min}} = 0.002$. We train the base model with $10\%$ condition dropout to enable CFG. The base model was trained for 5 days across 32 A100 GPUs with a batch size of 14 and learning rate of 1e-4 with Adam. For all score model experiments, we use CFG++ (Chung et al., 2024) with $w = 0.8$.

For **Presto-S**, following Yin et al. (2024) we use a fixed guidance scale of $w = 4.5$ throughout distillation for the teacher model as CFG++ is not applicable for the distribution matching gradient. We use 5 fake score model (and discriminator) updates per generator update, following Yin et al. (2024), as we found little change in performance when varying the quantity around 5 (though using $\leq 3$ updates resulted in large training instability). Note that throughout **Presto-S**, the fake score model and the discriminator share an optimizer state. Additionally, we use a learning rate of 5e-7 with Adam for both the generator and fake score model / discriminator. We set $\nu_1 = 0.01$ and $\nu_2 = 0.005$ following Yin et al. (2024). For all step distillation methods, we distill each model with a batch size of 80 across 16 Nvidia A100 GPUs for 32K iterations. We train all layer distillation methods for 60K iterations with a batch size of 12 across 16 A100 GPUs with a learning rate of 8e-5. For **Presto-L**, we set $\nu = 0.1$.

## A.2 EXPERIMENTAL DETAILS

### A.2.1 BASELINE DETAILS

Our benchmarks are divided into two main classes: acceleration algorithms and external open-source models. For acceleration algorithms, we distill our internal base model per method, utilizing publicly available code as a reference when available (Song et al., 2023; Saito et al., 2024; Yin et al.,

2024). For the open-source external models, we use the models directly in their default setups as recommended by Copet et al. (2023); Evans et al. (2024c).

- Consistency Models (CM) (Song et al., 2023; Bai et al., 2024): This distillation technique learns a mapping from anywhere on the diffusion process to the data distribution (i.e. $\boldsymbol{x}_t \to \boldsymbol{x}_0$) by enforcing the self-consistency property that $G_\phi(\boldsymbol{x}_t, t) = G_\phi(\boldsymbol{x}_{t'}, t') \quad \forall t, t'$. We follow the parameterization used in past audio works (Bai et al., 2024; Novack et al., 2024a) that additionally distills the CFG parameter into the model directly.
- SoundCTM (Saito et al., 2024): This approach distills a model into a consistency *trajectory* model (Kim et al., 2023) that enforces the self-consistency property, learning an anywhere-to-anywhere mapping. SoundCTM forgoes the original CTM adversarial loss and calculates the consistency loss via intermediate base model features.
- DITTO-CTM (Novack et al., 2024a), This audio approach is also based off of (Kim et al., 2023), yet brings the consistency loss back into the raw outputs and instead replaces CTM's multi-step teacher distillation with single-step teacher (like CMs) and removes the learned target timestep embedding, thus more efficient (though less complete) than SoundCTM.
- DMD-GAN (Yin et al., 2024): This approach removes the distribution matching loss from DMD2, making it a fully GAN-based finetuning method, which is in line with past adversarial distillation methods (Sauer et al., 2023)).
- ASE (Moon et al., 2024), This funetuning approach for diffusion models, as discussed in Sec. 3.3, finetunes the base model with the standard DSM loss, but for each noise level drops a fixed number of layers, starting at the back of the diffusion model's DiT blocks.
- MusicGen (Copet et al., 2023): MusicGen is a non-diffusion based music generation model that uses an autoregressive model to predict discrete audio tokens (Dfossez et al., 2022) at each timestep in sequence, and comes in small, medium, and large variants (all stereo).
- Stable Audio Open (Evans et al., 2024c): Stable Audio Open is a SOTA open-source audio diffusion model, which can generate variable lengths up to 45s in duration. Stable Audio Open follows a similar design to our base model, yet uses cross-attention for conditioning rather than AdaLN which we use, which increases runtime.

### A.2.2 METRICS DETAILS

We use Frechet Audio Distance (FAD) (Kilgour et al., 2018), Maximum Mean Discrepancy (MMD) (Jayasumana et al., 2024), and Contrastive Language-Audio Pretraining (CLAP) score (Wu et al., 2023), all with the CLAP-LAION music backbone (Wu et al., 2023) given its high correlation with human perception (Gui et al., 2024). FAD and MMD measure audio quality/realness with respect to Song Describer (lower better), and CLAP score measures prompt adherence (higher better). When comparing to other models, we also include density (measuring quality), recall and coverage (measuring diversity) (Naeem et al., 2020), and real-time factor (RTF) for both mono (M) and stereo (S, using MusicHiFi), which measures the total seconds of audio generated divided by the generation time, where higher is better for all.

## A.3 PRESTO-S ALGORITHM

---

**Algorithm 1 Presto-S**

---

**input** : generator $G_\phi$, real score model $\mu_{\text{real}}$, fake score model $\mu_\psi$, discriminator $D_\psi$, CFG weight $w$, $p_{\text{gen}}(\sigma^{\text{inf}})$, $p_{\text{DMD}}(\sigma^{\text{train}})$, $p_{\text{DSM}}(\sigma^{\text{train}})$, $p_{\text{GAN}}(\sigma^{\text{train}})$, real sample $x_{\text{real}}$, GAN weights $\nu_1, \nu_2$, optimizers $g_1, g_2$, weighting function $\lambda$

1: $\sigma \sim p_{\text{gen}}(\sigma^{\text{inf}})$
2: $\epsilon_{\text{gen}} \sim \mathcal{N}(0, I)$
3: $\hat{x}_{\text{gen}} = G_\phi(x_{\text{real}} + \sigma \epsilon_{\text{gen}}, \sigma)$
4: **if** generator turn **then**
5: $\quad \sigma \sim p_{\text{DMD}}(\sigma^{\text{train}})$
6: $\quad \epsilon_{\text{dmd}} \sim \mathcal{N}(0, I)$
7: $\quad \nabla_\phi \mathcal{L}_{\text{DMD}} = ((\mu_\psi(\hat{x}_{\text{gen}} + \sigma \epsilon_{\text{dmd}}, \sigma) - \tilde{\mu}_{\text{real}}^w(\hat{x}_{\text{gen}} + \sigma \epsilon_{\text{dmd}}, \sigma)) \cdot \nabla_\phi \hat{x}_{\text{gen}}$
8: $\quad \sigma \sim p_{\text{GAN}}(\sigma^{\text{train}})$
9: $\quad \epsilon_{\text{fake}} \sim \mathcal{N}(0, I)$
10: $\quad \mathcal{L}_{\text{GAN}} = \|1 - D_\psi(\hat{x}_{\text{gen}} + \sigma \epsilon_{\text{fake}}, \sigma)\|_2^2$
11: $\quad \phi \leftarrow \phi - g_1(\nabla_\phi \mathcal{L}_{\text{DMD}} + \nu_1 \nabla_\phi \mathcal{L}_{\text{GAN}})$
12: **else**
13: $\quad \sigma \sim p_{\text{DSM}}(\sigma^{\text{train}})$
14: $\quad \epsilon_{\text{dsm}} \sim \mathcal{N}(0, I)$
15: $\quad \mathcal{L}_{\text{fake-DSM}} = \lambda(\sigma) \|\hat{x}_{\text{gen}} - \mu_\psi(\hat{x}_{\text{gen}} + \sigma \epsilon_{\text{dsm}}, \sigma)\|_2^2$
16: $\quad \sigma_{\text{real}}, \sigma_{\text{fake}} \sim p_{\text{GAN}}(\sigma^{\text{train}})$
17: $\quad \epsilon_{\text{real}}, \epsilon_{\text{fake}} \sim \mathcal{N}(0, I)$
18: $\quad \mathcal{L}_{\text{GAN}} = \|D_\psi(\hat{x}_{\text{gen}} + \sigma_{\text{fake}} \epsilon_{\text{fake}}, \sigma_{\text{fake}})\|_2^2 + \|1 - D_\psi(x_{\text{real}} + \sigma_{\text{real}} \epsilon_{\text{real}}, \sigma_{\text{real}})\|_2^2$
19: $\quad \psi \leftarrow \psi - g_2(\nabla_\psi \mathcal{L}_{\text{fake-DSM}} + \nu_2 \nabla_\psi \mathcal{L}_{\text{GAN}})$
20: **end if**
**output** : $\phi, \psi$

---

We outline a condensed algorithm of **Presto-S** in math notation in Algorithm 1.

## A.4 PRESTO-S PSEUDO-CODE WALKTHROUGH

We provide a comprehensive algorithm walkthrough using PyTorch psuedo-code of our **Presto-S** training loop below. To perform **Presto-S**, we first define the corruption process for any given clean sample, according to either the training $p(\sigma^{\text{train}})$ or the inference $p(\sigma^{\text{inf}})$ noise distribution:

```
def diffuse(x, dist):
  eps = noise_normal_like(x)
  if dist == 'training':
    sigma = training_dist_like(x)
  elif dist == 'inference':
    sigma = inference_dist_like(x)
  return x + sigma * eps, sigma
```

We then define each of the component loss functions for the **Presto-S** continuous-time DMD2 distillation process. This corresponds to the three loss types: the distribution matching loss, the least-squares GAN loss, and the fake denoising score matching loss. For the distribution matching loss, we corrupt some generated sample according to the training distribution and then pass that into both the fake and real score models (where the real score model uses classifier-free guidance). The difference in these scores forms the distribution matching gradient:

```
def dmd(x, real_score_model, fake_score_model, cfg):
  x_noise, sigma = diffuse(x, 'training')
  fake_denoised = fake_score_model(x_noise, sigma)
  real_denoised = real_score_model(x_noise, sigma, cfg)
  return fake_denoised - real_denoised
```

For the least-squares GAN loss, we corrupt some sample (either real or generated) according to the training distribution and pass this through the discriminator (which itself involves first passing

through some of the fake score model to extract intermediate features). The output of the discriminator is then passed into the least-squares loss against some target value (i.e. the generator wants to push the discriminator outputs on generated samples towards 1, while the discriminator aims to push generated samples towards 0 and real samples towards 1):

```
1  def gan(x, discriminator, tgt=1):
2    x_noise, sigma = diffuse(x, 'training')
3    d_out = discriminator(x_noise, sigma)
4    return mse(tgt, d_out)
```

Finally, we have the fake DSM loss. This loss is identical to the normal diffusion loss (with a weighted MSE between the outputs of the score model and the clean data), yet will be calculated treating *generator* outputs as the ground truth clean data and using the fake score model:

```
1  def dsm(x, fake_score_model):
2    x_noise, sigma = diffuse(x, 'training')
3    x_denoised = fake_score_model(x_noise, sigma)
4    return weighted_mse(x, x_denoised, sigma)
```

Given these helper loss functions, we can now proceed with the main distillation loop, which is as follows. For both the generator and discriminator turns, we first corrupt some real input data according to the inference distribution, and pass this through our generator to get the generator outputs x_denoised (steps (1) and (4) in Fig. 8). If it is a generator turn (which happens once for every 5 fake score turns), we calculate the distribution matching loss (step (2)) and the generator adversarial loss (step (3)) on x_denoised and update the generator. If it is a fake score turn, we calculate and the discriminator's adversarial loss (step (5)) on both the generated x_denoised and real samples x and the fake DSM loss (step (6)) on x_denoised, thus updating the fake score model and the discriminator:

```
1  def forward(
2    x, generator, discriminator, fake_score_model, real_score_model,
      generator_turn, nu_1, nu_2
3  ):
4    # step (1) and (4)
5    x_noise, sigma = diffuse(x, 'inference')
6    x_denoised = generator(x_noise, sigma)
7
8    if generator_turn: # GENERATOR TURN
9      # Distribution Matching Loss, step (2)
10     dmd_loss = dmd(x_denoised, real_score_model, fake_score_model, cfg)
11
12     # Generator Adversarial Loss, step (3)
13     g_loss = gan(x_denoised, discriminator, 1)
14
15     loss = dmd_loss + nu_1 * g_loss
16   else: # FAKE SCORE TURN
17     # Discriminator Adversarial Loss, step (5)
18     d_loss = gan(x, discriminator, 1) + gan(x_denoised, discriminator, 0)
19
20     # fake DSM loss, step (6)
21     dsm_loss = dsm(x_denoised, fake_score_model)
22
23     loss = dsm_loss + nu_2 * d_loss
24   return loss
```

This constitutes one full update of the **Presto-S** process, alternating between the generator and fake score model / discriminator updates. At inference time, we can feed in pure noise and alternate between generating clean data with our generator and adding progressively smaller noise back to the generation (for some pre-defined list of noise levels), allowing for multi-step sampling:

```
1  def inference(generator, sigmas, start_noise):
2    x = start_noise
3    for sigma in sigmas:
4      x = x + noise_normal_like(x) * sigma
5      x = generator(x, sigma)
6    return x
```

## A.5 PRESTO-S EXPANDED DIAGRAM

For an in-depth visual illustration of **Presto-S**, please see Fig. 8 and Fig. 9 for expanded training and inference diagrams.

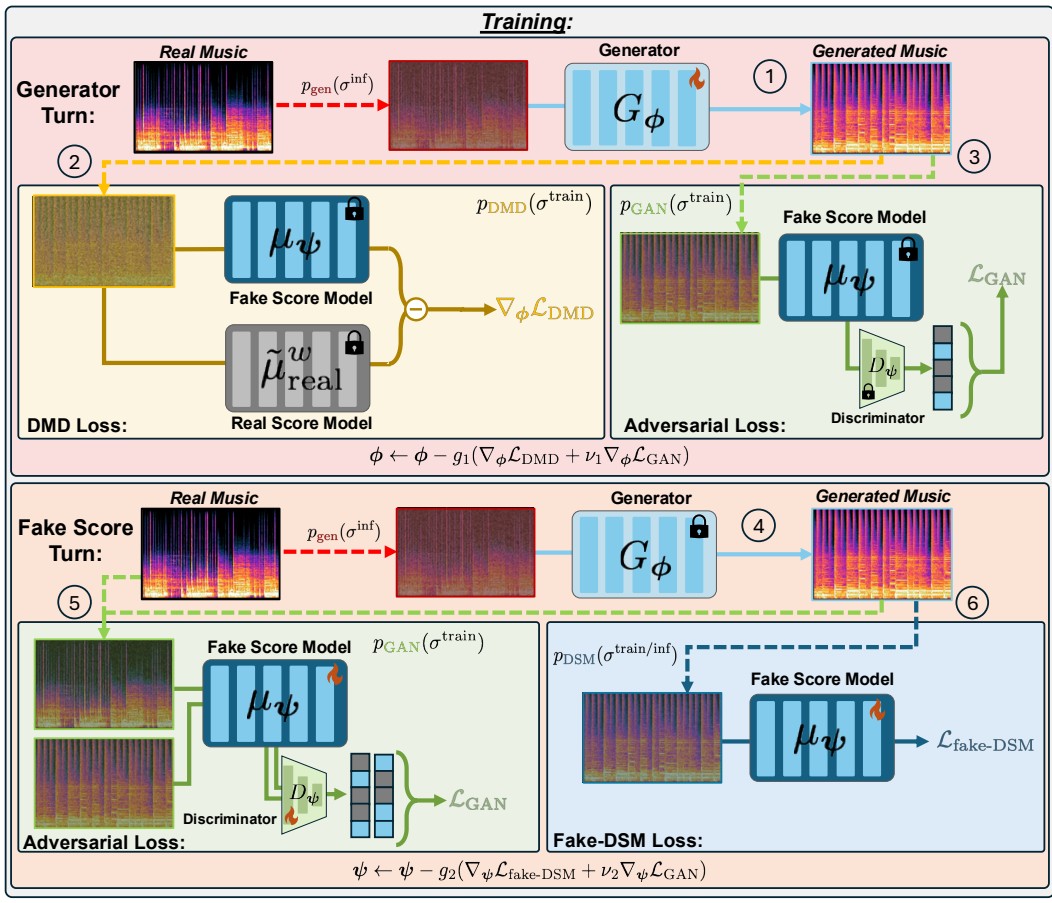

Figure 8: Presto-S training process.

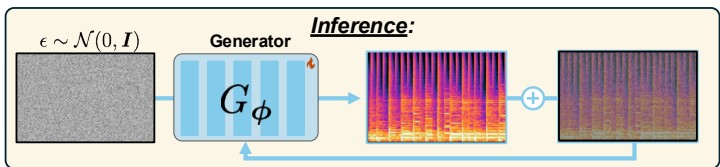

Figure 9: Presto-S inference. For multi-step sampling, we use ping-pong-like sampling.

A.6 **PRESTO-L** ALGORITHM

---

**Algorithm 2 Presto-L**

---

**input** : pre-trained score model $\mu_{\boldsymbol{\theta}}$, real sample $\boldsymbol{x}_{\text{real}}$, self-teacher weight $\nu$, optimizer $g, g_2$, weighting function $\lambda$, # of DiT blocks $B$, budget mapping $\ell$, layer drop function **LD**

1: $\sigma \sim p(\sigma^{\text{train}})$
2: $b = \ell(\sigma)$
3: $\epsilon \sim \mathcal{N}(0, \boldsymbol{I})$
4: $\hat{\boldsymbol{x}}_{\boldsymbol{L}}, h_{\boldsymbol{L}} = \textbf{LD}(\mu_{\boldsymbol{\theta}}, \boldsymbol{x}_{\text{real}} + \sigma\epsilon, \sigma, b)$
5: $\hat{\boldsymbol{x}}_{\text{full}}, h_{\text{full}} = \textbf{LD}(\mu_{\boldsymbol{\theta}}, \boldsymbol{x}_{\text{real}} + \sigma\epsilon, \sigma, B)$
6: $\mathcal{L}_{\text{DSM}} = \lambda(\sigma)\|\boldsymbol{x}_{\text{real}} - \hat{\boldsymbol{x}}_{\boldsymbol{L}}\|_2^2$
7: $\mathcal{L}_{\text{st}} = \|h_{\boldsymbol{L}} - \texttt{sg}(h_{\text{full}})\|_2^2$
8: $\boldsymbol{\theta} \leftarrow \boldsymbol{\theta} - g(\nabla_{\boldsymbol{\theta}}\mathcal{L}_{\text{DSM}} + \nu\nabla_{\boldsymbol{\theta}}\mathcal{L}_{\text{st}})$
**output** : $\boldsymbol{\theta}$

---

**Algorithm 3 LD**: Modified DiT forward pass with layer dropping and budget conditioning.

---

**input** : score model noise embedder $\mu_{\boldsymbol{\theta}}^{\text{noise}}$, score model budget embedder $\mu_{\boldsymbol{\theta}}^{\text{budget}}$, score model DiT blocks $\{\mu_{\boldsymbol{\theta}}^i\}_{i=1}^B$, score model budget AdaLN $\mu_{\boldsymbol{\theta}}^{\text{LN}}$, score model output layer $\mu_{\boldsymbol{\theta}}^{\text{final}}$, input $\boldsymbol{x}$, noise level $\sigma$, budget $b$

1: $\boldsymbol{e}_\sigma = \mu_{\boldsymbol{\theta}}^{\text{noise}}(\sigma)$                           // embed noise level
2: $\boldsymbol{e}_b = \mu_{\boldsymbol{\theta}}^{\text{budget}}(b)$                           // embed budget
3: $\boldsymbol{e} = \boldsymbol{e}_\sigma + \boldsymbol{e}_b$
4: **for** $i := 1$ to $b - 1$ **do**
5:   // apply first b−1 DiT blocks
6:   $x = \mu_{\boldsymbol{\theta}}^i(x, \boldsymbol{e})$
7: **end for**
8: $x = \mu_{\boldsymbol{\theta}}^B(x, \boldsymbol{e})$                           // apply final DiT block
9: $x = \mu_{\boldsymbol{\theta}}^{\text{LN}}(x, \boldsymbol{e}_b)$                           // apply budget−based AdaLN
10: $h = x/\|x\|_2$                           // get normalized hidden state for $\mathcal{L}_{\text{st}}$
**output** : $\mu_{\boldsymbol{\theta}}^{\text{final}}(x), h$

---

We show the full algorithm in detail for Presto-L in Algorithm 2, which proceeds as a modified version of standard diffusion training like in Moon et al. (2024). We first sample some noise level $\sigma$, and then map the noise level to its corresponding budget $b$ given some mapping function $\ell(\cdot)$. Following Moon et al. (2024), $\ell : \mathbb{R} \to \{i\}_{i=1}^B$ is a deterministic map from the percentile of the noise level according to the training noise distribution $F(\sigma)$ (where $F$ is the cumulative distribution function) to some budget amount, which we write as $[q_1, q_2, q_3, q_4, q_5]$ for a mapping based on descending *quintiles* (e.g. $q_1 = 14$ means that all noise levels in the largest quintile drop 14 layers).

We then call the modified forward function of the model **LD** (see Algorithm 3) on the noisy inputs with both the given budget $b$ and the full budget $B$ (i.e. using all DiT blocks). **LD** modifies the forward pass of the model by (1) adding a global budget embedding that is added to the noise embedding (2) only iterating through the first $b - 1$ DiT blocks followed by the final DiT block (to preserve final block behavior, see Section 3.3) (3) adding an additional AdaLN conditional only on the budget after the final DiT block and (4) also returning the normalized hidden state of the model (i.e. the input to the final layer of the DiT, normalized along the channel dimension). We calculate the standard denoising score matching loss $\mathcal{L}_{\text{DSM}}$ as normal but with our layer-dropped outputs, and additionally calculate $\mathcal{L}_{\text{st}}$ as the MSE between the layer-dropped hidden state and the full budget hidden state (with a stop-gradient operation on the full budget pass.

A.7 ANALYZING FAILURE MODES OF COMBINED LAYER AND STEP DISTILLATION

We empirically discovered a number of failure modes when trying to combine step and layer distillation. As noted in Section 4.6, the heavier per-layer requirements of distilled few-step generation made all standard dropping schedules (Moon et al., 2024) intractable and prone to quick generator collapse, necessitating a more conservative dropping schedule. In Fig. 10, we show the generator

loss, discriminator loss, distribution matching gradient, and the discriminator's accuracy for the *real* inputs over distillation, for a number of different setups:

Figure 10: Step distillation losses for early distillation for multiple combination methods. **Presto-LS** is the only setup that avoids generator degradation and high variance distribution matching gradients.

- **Presto-S**, pure step distillation mechanism (blue).
- **Presto-LS**, optimal combined setup where we pretrain the model with **Presto-L** and then perform **Presto-S**, but with keeping the real and fake score models initialized from the original score model (orange).
- LS with L-Fake/Real, which mimics **Presto-LS** but uses the **Presto-L** model for the fake and real score models as well (green).
- Step then Layer, where we first perform **Presto-S** distillation and then continue distillation with **Presto-L** layer dropping on the generator (red).
- Step and Layer jointly, where we perform **Presto-S** and **Presto-L** at the same time initialized from the original score model (purple),

We see that the runs which do not initialize with pretrained **Presto-L** (Step then Layer, Step and Layer) show clear signs of generator degradation, with increased generator loss, decreased discriminator loss, and notably near perfect accuracy on real samples, as attempting to learn to drop layers from scratch during step distillation gives strong signal to the discriminator. Additionally, LS with L-Fake/Real inherits similar collapse issues but has a higher variance distribution matching gradient as the layer-distilled real and fake score models are poor estimators of the gradient.

## A.8 INFERENCE-TIME NOISE SCHEDULE SENSITIVITY ANALYSIS

Given our final **Presto-LS** distilled 4-step generator, we show how changing the inference-time noise schedule can noticeably alter the outputs, motivating our idea of a continuous-time conditioning.

The EDM inference schedule follows the form of:

$$\sigma_{i<N} = \left( \sigma_{\max}^{1/\rho} + \frac{i}{N-1}(\sigma_{\min}^{1/\rho} - \sigma_{\max}^{1/\rho}) \right)^{\rho}, \qquad (9)$$

where increasing the $\rho$ parameter puts more weight on the low-noise, high-SNR regions of the diffusion process. In Fig. 11, we show a number of samples generated from **Presto-LS** with identical

conditions and latent codes (i.e. starting noise and all other added gaussian noise during sampling), only changing $\rho$, from the standard of 7 to 1000 (high weight in low-noise region). We expect further inference-time tuning of the noise schedule to be beneficial.

$$\rho = 7 \qquad\qquad \rho = 1000$$

*"A squirrel dancing in the backyard, uplifting" BPM=120*

*"song for my departed goldfish" BPM=80*

*"active winter on the mountains" BPM=100*

*"Epic videogame boss battle OST" BPM=140*

*"sea shanty for a drunken sailor" BPM=120*

Figure 11: Generations from **Presto-LS** from the *same* text prompt and latent code (i.e. starting noise and added noise during sampling), only varying the $\rho$ parameter between (7 and 1000). Purely shifting the noise schedule for 4-step sampling allows for perceptually distinct outputs.

## A.9 RTF ANALYSIS

We define the RTF for a model $\boldsymbol{\theta}$ as: $\mathrm{RTF}_b(\boldsymbol{\theta}) = \frac{bT_{\boldsymbol{\theta}}}{\mathrm{latency}_{\boldsymbol{\theta}}(b)}$, where $T_{\boldsymbol{\theta}}$ is the generation duration or how much *contiguous* audio the model can generate at once and latency$_{\boldsymbol{\theta}}(b)$ is the time it takes for generation following (Evans et al., 2024b; Zhu et al., 2024). This is different from the fixed-duration batched RTF used in Nistal et al. (2024). We test $b = 1$ as well as the *maximum* batch size we could attain for each model on a single A100 40GB to get a sense of maximum throughput. We show results in Table 3 and Table 4 for all components of our generative process, including latency metrics for generation (i.e. the diffusion model or distilled generator), decoding (i.e. VAE decoder from latents to audio) and the optional mono-to-stereo (M2S), as well as overall RTF/latency for mono and stereo inference. We omit the MusicGen models and the other step-distillation methods which

| Model | Generation Latency | Decoding Latency | Mono RTF | Mono Latency | M2S Latency | Stereo RTF | Stereo Latency |
|---|---|---|---|---|---|---|---|
| Stable Audio Open | 6159.01 | 887.99 | N/A | N/A | 0 | 4.54 | 7047 |
| Base DM | 4079.81 | 64.45 | 7.72 | 4144.27 | 205.31 | 7.36 | 4349.58 |
| ASE | 3200.73 | 64.45 | 9.80 | 3265.19 | 205.31 | 9.22 | 3470.50 |
| Presto-L | 3201.19 | 64.45 | 9.80 | 3265.64 | 205.31 | 9.22 | 3470.95 |
| SoundCTM | 238.06 | 64.45 | 105.78 | 302.51 | 205.31 | 63.01 | 507.83 |
| Presto-S | 204.98 | 64.45 | 118.77 | 269.43 | 205.31 | 67.41 | 474.74 |
| Presto-LS | 166.04 | 64.45 | 138.84 | 230.49 | 205.31 | 73.43 | 435.8 |

Table 3: Latency (ms) and real-time factor for a batch size of one on an A100 40GB GPU.

| Model | Generation Latency | Decoding Latency | Mono RTF | Mono Latency | M2S Latency | Stereo RTF | Stereo Latency |
|---|---|---|---|---|---|---|---|
| Stable Audio Open | 34602.86 | 4227.54 | N/A | N/A | 0 | 7.42 | 38830.4 |
| Base | 18935.26 | 1198.21 | 14.3 | 20133.46 | 1775.73 | 96.38 | 21909.19 |
| ASE | 14584.85 | 1198.21 | 18.25 | 15783.05 | 1775.73 | 96.25 | 17558.78 |
| Presto-L | 14655.02 | 1198.21 | 18.17 | 15853.23 | 1775.73 | 96.25 | 17628.96 |
| SoundCTM | 1135.65 | 1198.21 | 123.4 | 2333.86 | 1775.73 | 92.98 | 4109.58 |
| Presto-S | 715.41 | 1198.21 | 150.5 | 1913.62 | 1775.73 | 92.18 | 3689.34 |
| Presto-LS | 695.19 | 1198.21 | 152.11 | 1893.4 | 1775.73 | 92.13 | 3669.13 |

Table 4: Latency (ms) and real-time factor for max batch size on an A100 40GB GPU.

share the same RTF as **Presto-S**. For the fastest model **Presto-LS**, the biggest latency bottleneck is the mono-to-stereo model (Zhu et al., 2024) and VAE decoder. In future work, we hope to optimize the VAE and mono-to-stereo modules for faster inference.

## A.10 PRESTO-L DESIGN ABLATION

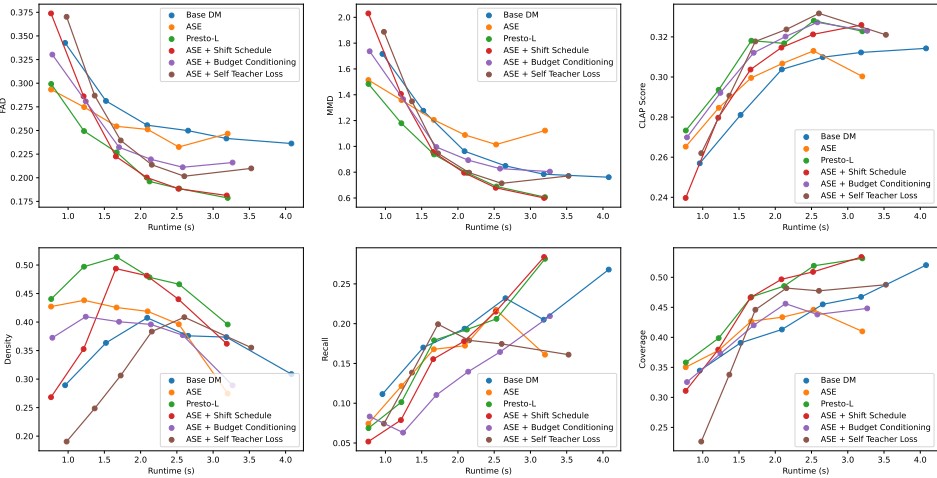

Figure 12: **Presto-L** ablation. Each individual change of our layer distillation vs ASE is beneficial.

To investigate how each facet of our **Presto-L** method contributes to its strong performance vs. ASE, we ran an additional ablation combining ASE with each component (i.e. the shifted dropping schedule, explicit budget conditioning, and the self-teacher loss). In Fig. 12, we see that the core of **Presto-L**'s improvements come from the shifted dropping schedule (which preserves final layer behavior), as the ASE+shift performs similarly to **Presto-L** on high-step FAD and MMD. Additionally, we find that the budget conditioning and self-teacher loss help text relevance more so than the shifted schedule does. All together, the combination of **Presto-L**'s design decisions leads to SOTA

audio quality (FAD/MMD/Density) and text relevance compared to any one facet combined with ASE.

## A.11 Discrete-Time Failure Modes

In Fig. 13, we visualize the poor performance of distilled models that use 1-2 step discrete-time conditioning signals. Notice that for the same random seed, the high-frequency performance is visually worse for discrete-time vs. continuous-time conditioning, motivating our proposed methods.

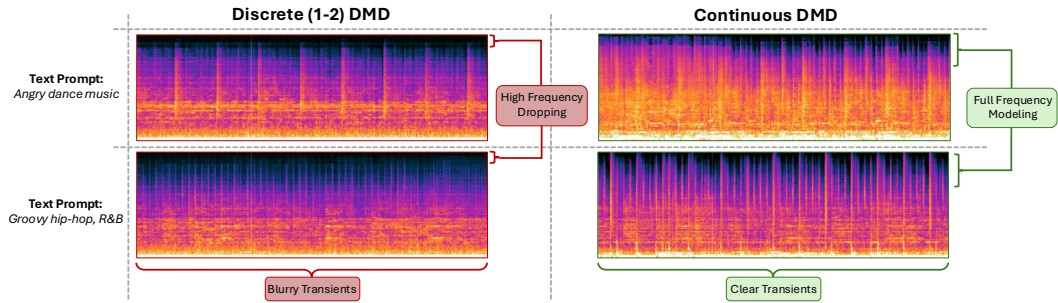

Figure 13: Failure mode of 1-2 step discrete models vs. continuous models (each row is same random seed and text prompt), with 2-step generation. Hip-Hop adjacent generations noticeably drop high frequency information, and render percussive transients (hi-hats, snare drums) poorly.

## A.12 Listening Test Results

We visualize our listening test results from Section 4.5 using a violin plot.

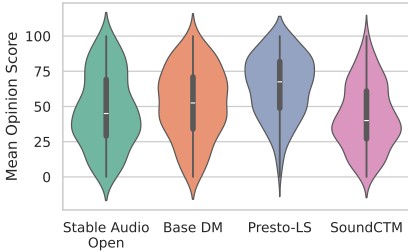

Figure 14: Violin plot from our listening test. Presto-LS is preferred over other baselines ($p < 0.05$).

## A.13 Rejection Sampling Results

We show rejection sampling results where we generate a batch during inference and then use CLAP to reject the $r$ least similar generations to the input text prompt. CLAP rejection sampling improves CLAP Score and maintains (and sometimes *improves*) FAD and MMD, but reduces diversity.

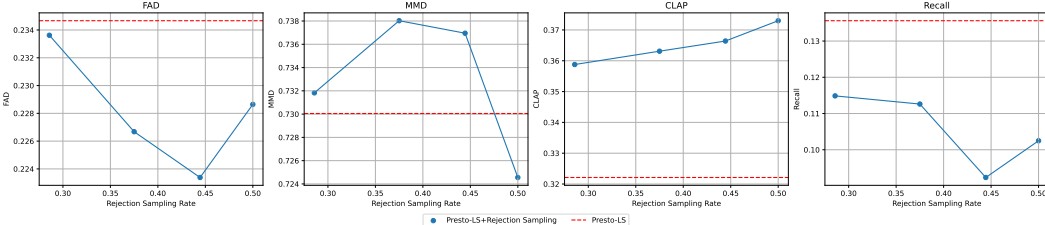

Figure 15: Rejection sampling eval metrics vs. rejection ratio. Base **Presto-LS** in red. CLAP rejection sampling improves both CLAP score and overall quality, while reducing diversity.

