# OpenReview forum: "Presto! Distilling Steps and Layers for Accelerating Music Generation"
_ICLR.cc/2025/Conference — ICLR 2025 Spotlight_

### Official Review · Reviewer_pB9s · 2024-11-03

**Soundness:** 4
**Presentation:** 3
**Contribution:** 3
**Rating:** 8
**Confidence:** 5

**Summary:**

The authors propose Presto, a distilled diffusion-based text-to-music model. Specifically, they introduce Presto-S, a model that applies DMD2 to a text-to-music generative model, and Presto-L, a model that performs layer-based distillation by extending ASE. Additionally, they propose Presto-LS, which combines these two distillation techniques. Experimental evaluations demonstrate that all three models achieve high-quality and efficient text-to-music generation.

**Strengths:**

- The authors propose the first diffusion-based text-to-music distillation model to successfully incorporate GAN loss.
- They also introduce a layer-distillation method to reduce computational cost at each sampling step, which has not yet been explored in the text-to-music task.
- Additionally, they present a methodology for integrating these two orthogonal distillation techniques.
- The effectiveness of these three methods is demonstrated through extensive evaluation.

**Weaknesses:**

- The explanation of Presto-L seems to assume prior knowledge of ASE, which is unfriendly for readers. For better readability and self-containment, it would be helpful to explain Presto-L more thoroughly by including an algorithm, equations, or an illustrative figure of the distillation process, as done for Presto-S. At the very least, a more detailed literature review of ASE should be provided. (ASE is a relatively new method presented at ICML 2024, and as the authors note in the last paragraph of Section 2, such layer-distillation methods have not yet been explored in text-to-music, underscoring the need for a detailed explanation)
- The contribution to the research community in terms of reproducibility is limited. Therefore, I strongly recommend either releasing the codebase or providing sufficient implementation details to facilitate straightforward replication, at least to allow future research to conduct follow-up studies (see also Questions).
    - While I understand from the reproducibility statement that the authors chose not to release checkpoints, it should still be feasible to release the codebase itself (as done in some prior works [1][2]) or to include sufficient implementation details for reproduction in the Appendix.

[1] Evans, Z., Carr, C.J., Taylor, J., Hawley, S.H. and Pons, J., 2024. Fast timing-conditioned latent audio diffusion. arXiv preprint arXiv:2402.04825.

[2] Evans, Z., Parker, J.D., Carr, C.J., Zukowski, Z., Taylor, J. and Pons, J., 2024. Long-form music generation with latent diffusion. arXiv preprint arXiv:2404.10301.

**Questions:**

- Around L90–L91 in Introduction, the authors mention TTA. Are there plans to conduct text-to-audio generation experiments during the rebuttal period?
   - If not, claiming it as a contribution in the Introduction is an overstatement.
- How exactly is the CFG-augmented real score, represented by $\tilde{\mu}^{w}_{\text{real}}$ in Eq (5), formulated?
     - Is it something like $\mu_{\text{real}}( \hat{x}_{\text{gen}}+\sigma\epsilon, \sigma, \varnothing) + w (\mu_{\text{real}}( \hat{x}_{\text{gen}}+\sigma\epsilon, \sigma, \bf{c}) - \mu_{\text{real}}( \hat{x}_{\text{gen}}+\sigma\epsilon, \sigma, \varnothing)$, $where $\bf{c}$ is condition?
- Section 3.1, L210–L211: Regarding “the fake score model is updated 5 times as often as the generator to stabilize the estimation of the generator’s distribution,” how robust is this to different model training setups, architectures, and datasets? For example, if the number “5” were changed to “4” or “6,” how significantly would it affect training stability and generation quality in those sense?
- Section 3.1, L212–L214: In “Specifically, a discriminator head $D_{\psi}$ is attached to the intermediate feature activations of the fake score network,” which specific part of the DiT block do the “intermediate feature activations” refer to? Also, how is the architecture of the discriminator head $D_{\psi}$ actually configured? The explanation in Section 3.2.3 is still high-level for both points. Could you provide further detail using pseudo-code or a figure illustrating the architecture?
- In Algorithm 1, $\nu_1, \nu_2, g_1, g_2$ are mentioned. What values do the authors set as $\nu_1, \nu_2$? Are both $g_1, g_2$ optimized with Adam at a learning rate of 1e-4? Also, do the authors use learning rate schedulers?
- How is Presto-L distilled? The explanation in Section 3.3 and $\mathcal{L}_{\text{st}}$ alone seem insufficient to clarify the distillation process. Could you provide a more detailed explanation using an algorithm table, pseudo-code, or a figure illustrating the distillation process?
- Since multi-step sampling uses stochastic sampling like CM, I would expect that as the number of sampling steps increases, sample quality will eventually plateau and then degrade if increased further. Have you observed such a phenomenon when increasing the number of sampling steps to around 15, for example?

---

> ### Author Response · Authors · 2024-11-21
>
> We thank the reviewer for their review, and sincerely appreciate their detailed comments. Below, we address overall comments and concerns brought up in the review:
>
> >**”The explanation of Presto-L seems to assume prior knowledge of ASE…”**
>
> Regarding the noted weakness of assuming prior knowledge of ASE, we acknowledge and agree that our readability and self-containment can be improved. To address this, we have included pseudo-code for the Presto-L algorithm and a longer form description in Appendix A.6.
>
> >**”The contribution to the research community in terms of reproducibility is limited…I strongly recommend either releasing the codebase or providing sufficient implementation details“**
>
> We acknowledge this shortcoming and agree that releasing code (model checkpoints and data) would greatly help reproducibility as well as alleviate issues w.r.t. details and clarity. However, we do not plan to do so due to constraints outside of our control. Thus, to ameliorate this important issue and improve reproducibility of our work, we have followed your suggestion and included considerable additions in the paper as we note in our general rebuttal. To expand more here, we have
> 1) Added pseudo-code and long-form explanation for our layer distillation method, Presto-L (Appendix A.6)
> 2) Added detailed torch-style pseudo-code walkthrough for Presto-S (Appendix A.4) to complement our existing pseudo-code (Appendix A.3 Algorithm 1)
> 3) Added a new tutorial figure that expands upon figure 1 to provide a careful and in-depth system overview to better describe the high-level method design (Appendix A.5). Please also see our main rebuttal.
>
> >**”Are there plans to conduct text-to-audio generation experiments during the rebuttal period?”**
>
> We appreciate the reviewer catching this error on our part, and have since removed the mention of TTA, as we only focus on the subset task of TTM and do not plan to run general TTA experiments during this period.
>
> >**”How exactly is the CFG-augmented real score, represented by  $\mu_{\text{real}}^w$  in Eq (5), formulated?”**
>
> The reviewer is exactly correct, as $\mu_{\text{real}}^w$ is calculated as:
>
> $\tilde{\mu}^w_{\theta}(x, \sigma, e) = \mu_{\theta }(x, \sigma,\emptyset) + w (\mu_{\theta}(x, \sigma, e) - \mu_{\theta }(x, \sigma, \emptyset))$
>
> Where $e$ is the condition embedding. This is noted as an inline equation on line 147.
>
> >**”how robust is this to different model training setups, architectures, and datasets? For example, if the number “5” were changed to “4” or “6,” how significantly would it affect training stability and generation quality in those sense?”**
>
> Though we followed the recipe of DMD2 with 5 fake score / discriminator updates per generator update, initial experiments showed that varying the update rate in the [4, 6] range did not impact performance much. When we reduce the update rate $\le3$, the performance noticeably worsened, and in particular led to poor estimation of the fake score distribution. We have since added clarifying information in Appendix A.1 to discuss this fact.
>
> >**”which specific part of the DiT block do the “intermediate feature activations” refer to? Also, how is the architecture of the discriminator head $D_\psi$ actually configured?”**
>
> We recognize the reviewer's concern, and have since updated the appendix to highlight more details on the Discriminator design. Specifically, our discriminator design follows the design of the original DMD2 work, where our discriminator is a series of 4 convolution blocks interleaved with groupnorm and SiLU activations, with a final linear layer to collapse to a single channel, using the following modifications:
> - As our data representation are 1D sequences, all 2D convolutions are replaced with 1D convolutions
> - Since we use the LSGAN formulation, we do not downsample our representation all the way down to a single true / false value. Instead, the output of the discriminator is a heavily downsampled version of the input representation, specifically at around 2.8Hz.
> - Since our model is DiT based rather than a UNet as in DMD2, the “intermediate feature activations” refer to the outputs of the 14th DiT block of the model (i.e. the halfway point of our 28 block DiT), as DiTs lack a real “bottleneck” layer (that UNets have).
> We have thus added this information into Appendix A.1 to actively highlight this.
>
> >**”What values do the authors set as $\nu_1, \nu_2$ ? Are both $g_1, g_2$ optimized with Adam at a learning rate of 1e-4? Also, do the authors use learning rate schedulers?”**
>
> We thank the reviewer for catching this lack of information on our part, and have thus modified Appendix A.1 to provide full details. Specifically, for both $g_1, g_2$ we use Adam with a learning rate of 5e-7. We set $\nu_1=0.01$ and $\nu_2=0.005$. Following DMD2, we do not use any learning rate schedulers.

---

> > ### Author Response · Authors · 2024-11-21
> >
> > >**”Since multi-step sampling uses stochastic sampling like CM, I would expect that as the number of sampling steps increases, sample quality will eventually plateau and then degrade if increased further.”**
> >
> > The reviewer is right in their intuition: overall we found that performance generally plateaued around 4 steps, and started degrading performance even at 8 steps of the model. Based on the reviewer’s suggestion, we have generated performance metrics for both 8 and 16 steps (as shown below), and see the expected performance degradation due to the stochastic multi-step sampling:
> > | # Sampling Steps | FAD  | MMD  | CLAP  | Density | Recall | Coverage |
> > |-|-|-|-|-|-|-|
> > | 8 | 0.27 | 0.95 | 33.78 | 0.35 | 0.07 | 0.25 |
> > | 16 | 0.32 | 1.48 | 32.18 | 0.27 | 0.02 | 0.17 |
> >
> > In future work, we hope to discover whether there are similar formulations of Presto-S that allow for more stable and predictably scalable multi-step sampling.

---

> ### Comment · Reviewer_pB9s · 2024-11-22
>
> I sincerely appreciate the time and effort you put into thoroughly addressing my concerns. I now feel that all of them have been addressed very effectively. Thank you for presenting such an interesting paper—it was truly a pleasure to review your work.

---

### Official Review · Reviewer_eJot · 2024-11-03

**Soundness:** 4
**Presentation:** 3
**Contribution:** 3
**Rating:** 8
**Confidence:** 4

**Summary:**

This paper presents a distillation method to develop a fast music generation model. The authors modify the [DMD2](https://arxiv.org/abs/2405.14867) and [ASE](https://proceedings.mlr.press/v235/moon24a.html) frameworks to improve the model performance. The proposed method not only enables a model to generate music signals with a few NFEs but also makes the model size smaller. The authors demonstrated that their trained model outperforms models trained/equipped with previous methods in terms of multiple evaluation metrics, using their in-house data for training and the Song Describer dataset for evaluation. They also conducted comprehensive ablation studies to show that their choices are reasonable.

**Strengths:**

1. The paper is well written. I could understand their motivation, the proposed method, and the experimental results.
2. The authors modified the existing DMD2 and ASE frameworks while conducting ablation studies. As ablation studies are comprehensive, readers can understand the reasons for the modifications/choices.
3. The authors demonstrated that the proposed distillation method works well in several aspects (AD, MMD, CLAP scores, and subjective evaluation) despite its fast generation.

**Weaknesses:**

I do not think that the paper has a critical weak point, but let me mention some weaknesses.

- Although the paper is well-written, some readers would struggle to reproduce the results without their code public. (I understand some institutes/companies do not make their code publicly available due to their policy.)
- Since the proposed training framework is general, I thought I would like to see experimental results in the text-to-audio generation task evaluated on AudioCaps. Furthermore, if we have a DiT-based baseline model for text-to-image or text-to-video generation, we can apply the proposed method to them. I thought I would like to see the performance in those tasks as well.
- The explanation of baselines in Section 4.1 is a little unclear to me. My understanding is that the authors trained the teacher model and applied the proposed technique and previous ones (CM, SoundCTM, DITTO-CTM, DMD-GAN, and ASE). For MusicGen and Stable Audio Open, they just downloaded and evaluated the official distributed models. Whether my understanding is correct or not, a clearer description would be appreciated.

**Questions:**

I would appreciate the authors' response to my comment in "Weaknesses".

---

> ### Author Response · Authors · 2024-11-21
>
> We thank the reviewer for their detailed review, and are glad to see the recommendation of acceptance. Beyond our main rebuttal comments, please see our response to your individual comments below.
>
> >**”Although the paper is well-written, some readers would struggle to reproduce the results without their code public. (I understand some institutes/companies do not make their code publicly available due to their policy.)”**
>
> We acknowledge this shortcoming and agree that releasing code (model checkpoints and data) would greatly help reproducibility as well as alleviate issues w.r.t. details and clarity. Unfortunately, however, we do not plan to do so due to constraints outside of our control. To ameliorate the important issue and improve reproducibility of our work, we have gone to great lengths to address as we note in our general rebuttal. To expand more here, we have
> 1) Added pseudo-code for our layer distillation method, Presto-L
> 2) Added a detailed torch-style pseudo-code walkthrough for Presto-S to complement our existing pseudo-code (Appendix A.3 Algorithm 1)
> 3) Added a new walkthrough figure that expands upon figure 1 to provide a careful and in-depth system overview to better describe the high-level method design. Please also see our main rebuttal.
>
> >**”Since the proposed training framework is general, I thought I would like to see experimental results in the text-to-audio generation task evaluated on AudioCaps. Furthermore, if we have a DiT-based baseline model for text-to-image or text-to-video generation, we can apply the proposed method to them.”**
>
> Thank you for the important comment and desire to apply our proposed method on other domains such as general-purpose audio generation, image generation, and video generation. While we would be very motivated to include such work, we believe expanding the scope to this level would be relatively difficult to contain within our manuscript here and still manage to focus on our goal of music generation.
>
> >**”The explanation of baselines in Section 4.1 is a little unclear to me. My understanding is that the authors trained the teacher model and applied the proposed technique and previous ones... For MusicGen and Stable Audio Open, they just downloaded and evaluated the official distributed models.”**
>
> We recognize the reviewer's concern, and the reviewer is exactly right in their idea of what Section 4.1 implies, with the acceleration algorithms being used on our base model and the external baselines just being used as is from the official repositories. To make this fact more clear, we have modified Section 4.1 slightly in order to highlight this difference and elaborated much more in Appendix A.2 to highlight this fact explicitly.

---

> > ### Comment · Reviewer_eJot · 2024-11-23
> > **Response to Authors**
> >
> > I sincerely appreciate your careful response to the reviewers' comments. The added figure and pseudo-code are helpful. On the other hand, I still think that an experimental result on general-purpose text-to-audio generation would be informative to readers because they can easily compare their new method with the Presto framework (if a Presto model is trained on public datasets such as AudioCaps and/or AudioSet) and because such an experiment will demonstrate the generalizability of this framework. However, I think this paper is already good enough for acceptance. I recommend the paper. It was a pleasure to review this great paper.

---

### Official Review · Reviewer_ZS9n · 2024-11-03

**Soundness:** 3
**Presentation:** 2
**Contribution:** 2
**Rating:** 5
**Confidence:** 4

**Summary:**

This paper introduces an innovative approach to improve text-to-music generative models by significantly reducing inference time. Focusing on score-based diffusion models, the authors present two main contributions: first, a reformulated few-step synthesis approach utilizing online GAN-based adversarial distillation, adapted specifically for continuous time diffusion through the DMD2 framework; and second, a layer distillation technique, inspired by ASE, which uses a layer-budgeting module to prioritize layers based on noise levels.

They further explore the combination of step and layer distillation, demonstrating cases where the methods complement each other effectively. Their experiments, conducted on latent diffusion models with VAE and DiT blocks, use internal datasets for training and Song Describer for evaluation. Comparative analysis on the CLAP-LAION dataset with FAD scores shows that each method, independently and in combination, improves upon alternatives while maintaining real-time factor (RTF) performance.

**Strengths:**

This paper marks an important step toward a unified approach for faster generative models by bridging adversarial and diffusion methods for text-to-music tasks. In particular, Section 3.1 demonstrates this by applying GAN-based distillation to continuous-time score-based diffusion models.

Aligning most distributions with the training distribution, as shown in Table 1, empirically improves all metrics, underscoring the robustness of their approach.

The authors also navigate the task of combining layer and step distillation, finding a highly effective recipe that substantially enhances generation speed.

Achieving a 10-18x speedup in audio generation is a noteworthy accomplishment, positioning this collection of methods as a valuable resource for practical applications in the field.

**Weaknesses:**

While the paper is well-structured in a general sense, certain sections are hard to follow due to the level of detail. For example, Figure 1 is dense and, despite a detailed caption, remains challenging to interpret. The figure also uses various notations that lack prior explanation, making it difficult for readers to follow the intended process.

Additionally, the models are trained on internally licensed data, so the results cannot be easily referenced in future work. With no indication that the code will be released, reproducing this complex setup—including multiple models, distributions, and training phases—would be exceptionally difficult. This limitation means future researchers may be restricted to evaluating audio examples alone rather than building upon this work directly.

**Questions:**

In line 35, does 5-20 seconds latency refer to 32 seconds audio? If yes, please clarify because it's mentioned in the abstract but not in the introduction.

---

> ### Author Response · Authors · 2024-11-21
>
> We thank the reviewer for the detailed review and are grateful for your critical feedback. Below, we address overall comments and concerns:
>
> >**”Figure 1 is dense and, despite a detailed caption, remains challenging to interpret. The figure also uses various notations that lack prior explanation, making it difficult for readers to follow the intended process.”**
>
> We recognize the density of figure 1, and have sought to address your feedback by improving readability. Specifically, we have added a further detailed diagram in Appendix A.5 (now Figures 8 and 9) that goes through the Presto-S process step-by-step. We also introduce multiple additional pseudo-code algorithms to alleviate any issues with respect to detail.
>
> >**”Additionally, the models are trained on internally licensed data, so the results cannot be easily referenced in future work. With no indication that the code will be released, reproducing this complex setup—including multiple models, distributions, and training phases—would be exceptionally difficult.”**
>
> Thank you for your concern and comments and acknowledge the difficulty in reproducing work, particularly when trained on licensed data to do our best to avoid copyright issues. To help address this, we have added multiple additional algorithm text blocks including a torch-style pseudo-code walkthrough for Presto-S to complement our existing pseudo-code and pseudo-code for Presto-L with corresponding textual explanation. For more discussion on the topic, please also see our main rebuttal as well as our ethics and reproducibility statement where we outlined our constraints w.r.t. open source code.
>
> >**”In line 35, does 5-20 seconds latency refer to 32 seconds audio? If yes, please clarify because it's mentioned in the abstract but not in the introduction.”**
>
> We thank the reviewer for catching this, and have since modified the introduction accordingly.

---

> > ### Author Response · Authors · 2024-12-02
> >
> > We kindly look forward to reviewer ZS9n’s response to our updated draft and rebuttal comments, as the window for discussion closes tomorrow.

---

### Official Review · Reviewer_Dv9d · 2024-11-04

**Soundness:** 4
**Presentation:** 4
**Contribution:** 4
**Rating:** 8
**Confidence:** 4

**Summary:**

The paper proposes "Presto!" for effective and efficient music generation. More specificly, Presto! is a set of model distillation techniques that aims at improving the inference efficiency of continuous diffusion models from two aspects:
1. overall inference steps,
2. runtime of each inference step.

The motivations and approaches are sound, and the results on the demo page are extremely convincing. Huge efforts have been done to applying latest SOTA methods to their own DNN architecture. Experiments are well designed to support the claims and motivations.

It is unfortunate that the authors do not plan to open their method, as I feel the framework is a little bit complex: it invovles extensive grid seearch for a better stability in adversarial training.

**Strengths:**

1. Very solid target: enhancing efficiency from both the number of steps and the runtime of a single ste.
2. novel to audio community to combine both step distillation and layer distillation, on top of the successful usage of an adversarial loss.
3. Clear presentation. The origin of the proposed techniques, and the differences between Presto! and prior arts are well explained. Related works are very up-to-date hence informative.
4. Great objective measurements. Presto! could outperform or keep on-par with the teacher model, while reducing the sampling steps to as few as 4, for the first time in audio community. The proposed continuous step distillation offers a more predictable performance gain when more inference budget is given.
5. Excellent subjective quality. The violin plot shows that Presto! outperforms Stable Audio Open and SoundCTM, two works published earlier this year, which is impressive. The samples in demo page sound great.

**Weaknesses:**

The ablation study indicates quite high complexity if we want to stably and effectively use Presto!. I am interested in how well the framework could be when applied to other VAEs or generator backbones, especially if no resource is open.

**Questions:**

Is it possible to open the implementation without sharing pretrained model weights? Or open a part of key tricks or components in the work?

---

> ### Author Response · Authors · 2024-11-21
>
> We thank the reviewer for their insightful review. Below we address comments brought up in the review:
>
> >**”I am interested in how well the framework could be when applied to other VAEs or generator backbones, especially if no resource is open. Is it possible to open the implementation without sharing pretrained model weights? Or open a part of key tricks or components in the work?”**
>
> We fully recognize the reviewer’s concern about the complexity and how it relates to open source code. To address this issue, please kindly see our main rebuttal. To expand here – we have taken several steps to improve the reproducibility and understanding of our work including adding pseudo-code for Presto-L as well as a more detailed torch-style pseudo code walkthrough for Presto-S to complement our existing algorithm. We would also like to note the original DMD2 paper [Yin 2024] does have open source code, so although the algorithm is different and applied to discrete-time diffusion models and not continuous-time models, it is closely related and can be used together with our torch-style pseudo code. It is our best intention that this will aid researchers in reproducing our method.
>
> Regarding other VAEs or generator backbones, in initial experiments we tried our method on variants of the final VAE and generator setup (such as changes in the compression rate of the VAE or the size of the DiT hidden state) and found similar results.

---

### Author Response · Authors · 2024-11-21
**Overall Rebuttal**

We would sincerely like to thank all the reviewers for their insightful comments and constructive feedback. We are thankful to acknowledge that most reviewers recommend acceptance as well as the reviewers highlighting overall clarity (Dv9d, eJot), methodological contributions (Dv9d, ZS9n, pB9s), and breadth of experimental validation (Dv9d, eJot, pB9s).

We note the single concern common among all reviewers is the lack of open source code (Dv9d, ZS9n, eJot, pB9s). Reviewers note that this issue impacts both reproducibility as well as clarity on details that would otherwise be easier to address. We acknowledge this shortcoming and agree that releasing code (model checkpoints and data) would greatly help reproducibility as well as alleviate issues w.r.t. details and clarity. Unfortunately, however, we do not plan to do so due to constraints outside of our control.

To thus ameliorate the important issue and improve reproducibility of our work, we have added a number of extensions to the paper in order to address this. More specifically, we:

1) Added a detailed walkthrough of Presto-S with torch-style pseudo-code for a new Appendix A.4 to complement our existing pseudo-code (Appendix A.3).
2) Added a new tutorial figure that expands upon figure 1 to provide a careful and in-depth system overview to better describe the high-level method in a new Appendix A.5.
3) Added pseudo-code and a detailed textual overview for our layer distillation method, Presto-L for a new Appendix A.6.

We believe that our torch-style pseudo-code (suggested by Dv9d and pB9s), in particular, will aid ICLR practitioners in reproducing our method and also help resolve any clarity issues that arise w.r.t. details.

---

### Author Response · Authors · 2024-11-27

We thank reviewers Dv9d, eJot, and pB9s for their response to our rebuttal and confirmation of their scores. As the period for updating revised PDFs ends soon, we look forward to reviewer ZS9n’s comments to our rebuttal.

---

### Meta-Review · Area_Chair_9qAU · 2024-12-21

**Metareview:**

**Paper Summary:**

This paper improves the inference speed of text-to-music diffusion models, adapting techniques for model distillation (DMD2) and inference acceleration (ASE) from prior work on vision. These adaptations are well-motivated, and experiments convincingly validate their high quality and low inference costs of the resulting models.

**Strengths:**

The reviewers praise the paper’s clarity, thoroughness, and convincing experiments. Reviewer eJot states “It was a pleasure to review this great paper” and reviewer pB9s similarly comments “it was truly a pleasure to review your work.” Even the most critical review, provided by reviewer ZS9n states "This paper marks an important step toward a unified approach for faster generative models by bridging adversarial and diffusion methods for text-to-music tasks.”

**Weaknesses:**

Every reviewer criticized the authors’ choice to withhold code and/or model checkpoints. I share this criticism. That said, the authors have taken significant steps during the rebuttal period to provide additional details (short of code) on the implementation of their methods.

**Conclusion:**

This work represents a substantial methodological contribution to the text-to-music literature. In my opinion, the concerns about reproducibility due to lack of code release have been adequately addressed by the additional descriptive material added to the appendix during the discussion period. I recommend acceptance.

**Additional Comments On Reviewer Discussion:**

See main review comments about code/weight release.

---

### Decision · Program_Chairs · 2025-01-22

Accept (Spotlight)